# A sparse quantized hopfield network for online-continual memory

Nicholas Alonso [1] & Jeffrey L. Krichmar [1,2]

An important difference between brains and deep neural networks is the way they learn. Nervous systems learn online where a stream of noisy data points are presented in a non-independent, identically distributed way. Further, synaptic plasticity in the brain depends only on information local to synapses. Deep networks, on the other hand, typically use non-local learning algorithms and are trained in an offline, non-noisy, independent, identically distributed setting. Understanding how neural networks learn under the same constraints as the brain is an open problem for neuroscience and neuromorphic computing. A standard approach to this problem has yet to be established. In this paper, we propose that discrete graphical models that learn via an online maximum a posteriori learning algorithm could provide such an approach. We implement this kind of model in a neural network called the Sparse Quantized Hopfield Network. We show our model outperforms state-of-the-art neural networks on associative memory tasks, outperforms these networks in online, continual settings, learns efficiently with noisy inputs, and is better than baselines on an episodic memory task.

A fundamental question in computational neuroscience and neuromorphic computing is the question of how to train deep neural networks using only local learning rules in the learning scenario faced by the brain, where data is noisy and presented in an online-continual fashion. Local learning rules use only information spatially and temporally adjacent to the synapse at the time of the update (e.g., pre-synaptic and post-synaptic neuron activity). Online-continual learning occurs when a stream of single data points are presented during training in a non-independent and identically distributed (non-i.i.d.) fashion (e.g., several data sets are presented one dataset at a time). Brains must learn under these conditions, and neuromorphic hardware embedded in real-world systems have similar constraints[1].

Standard approaches to deep learning have not provided a solution to this problem. The standard approach trains neural networks with stochastic gradient descent (SGD) implemented by the back-propagation algorithm (BP)[2]. BP is a non-local learning algorithm generally considered biologically implausible[3–5] and is difficult to make compatible with neuromorphic hardware[6,7]. Further, the standard training paradigm is offline learning, where data is mini-batched, i.i.d.,

non-noisy, and can be passed over for multiple epochs during training. Learning in the noisy, online-continual scenario is much more difficult than in the offline scenario. Unlike offline learners, online-continual learners must avoid problems like catastrophic forgetting and are pressured to deal with noise and to be more sample efficient (faster learners).

Furthermore, recent work on this problem has not approached all the aspects of the problem simultaneously. For example, although bio-plausible algorithms have been developed for deep networks, these algorithms are typically tested and developed for offline settings (e.g., refs. 8–11). Although progress has been made on online-continual learning, essentially all of this work uses BP in some capacity (see refs. 12–17). Some works do test local learning algorithms in online and continual settings separately, but do not address the online and con-tinual setting simultaneously or focus on shallow recurrent networks (e.g., refs. 18–20). Therefore, local learning models that perform online-continual learning without resorting to BP are needed.

We attempt to remedy this situation by making the following contributions: (1) Unlike previous works on online-continual learning,

[1]Department of Cognitive Science, University of California, Irvine, CA, USA. [2]Department Computer Science, University of California, Irvine, CA, USA.
✉ e-mail: nalonso2@uci.edu

which tend to focus on classification, we study the more general and basic task of associative memory, i.e., the basic process of storing and retrieving corrupted and partial patterns. We believe studying this basic task could yield ideas that apply across a wide range of tasks, rather than a single narrow task, like classification. (2) We propose a general approach to online-continual associative memory, based on the idea that a sparse neural code, representing the value of a discrete variable, can deal with both noisy and partial input and prevent catastrophic forgetting. We propose implementing this strategy using discrete graphical models that learn via algorithms similar to maximum a posteriori (MAP) learning, where MAP learning has the advantage of using local learning rules. (3) We implement this approach in a neural network called the sparse quantized Hopfield network (SQHN), an energy-based model that optimizes an energy function and utilizes a learning algorithm that combines neuro-genesis (neuron growth) and local learning rules, both engineered specifically to yield high performance in the noisy and online-continual setting. (4) We develop two memory tasks, which are new to the recent machine learning literature on associative memory models, the noisy encoding task and an episodic memory task. 5) We run a variety of tests showing that SQHN significantly outperforms baselines on these new tasks and matches or exceeds state-of-the-art (SoTA) on more standard associative memory tasks.

## Results

### Toward a foundation for local, online-continual memory models

Our goal is to design a model that can (1) deal with noisy, partial inputs in associative memory tasks, (2) learn in a sample-efficient way that avoids catastrophic forgetting in online-continual learning scenarios, and (3) use only local learning rules. Our proposed approach has three parts.

First, we propose that using neural networks which utilize quantized neural codes could provide a principled approach to associative recall. More specifically, we imagine a general process, akin to vector quantization[21], which maps continuous-valued input vectors, to a neural code with a finite set of values. Such a mapping is necessarily a process of pattern completion, i.e., a process where a subset of distinct vectors are mapped to the same vector/code. The associative memory problem may also be cast as one of pattern completion, where the goal is to reconstruct stored data points, $x$, given corrupted or partial versions, $\tilde{x}$, of it, i.e., $[\tilde{x}_0, \tilde{x}_1, \ldots] \rightarrow x$. A general process akin to vector quantization can be used to perform this mapping via $[\tilde{x}_0, \tilde{x}_1, \ldots] \rightarrow_\theta h^* \rightarrow_\theta x$, where $h^*$ is a single discrete neural code determined by parameters $\theta$ and the inputs. Pattern completion is intuitively more difficult with continuous latent codes, since these codes may vary in an infinite number of ways, making it more difficult to map many corrupted versions of a data point to the same latent code and reconstruction (e.g., see experiments and discussion).

Second, we primarily use parameter isolation to avoid catastrophic forgetting, which is a strategy that has recent success in BP-based deep learning models (e.g., refs. [22–25]). This strategy allocates subsets of new and old parameters to different tasks during training, as needed. By only using and updating small subsets of parameters each iteration, models are able to drastically avoid forgetting. However, there needs to be a principled method to decide which parameters to update or add at which times.

Third, we propose using a MAP learning algorithm as a local learning algorithm, in discrete-graphical models, which naturally implement vector quantization and parameter isolation. MAP learning works by first performing inference over hidden variables to find their specific values, $h^*$, that maximize the posterior $P(h^*|x, \theta) \propto P(h^*, x|\theta)$. In discrete graphical models, these values are integers. The process of vector quantization maybe seen as a MAP inference process, where inputs are assigned the component with the highest posterior probability. Parameters are then updated to further increase the probability

of the joint $P(h^*, x|\theta)$. Local learning rules are naturally used in this algorithm. Further, if integer values are represented by sparse, one-hot vectors, updates are also sparse, i.e., perform parameter isolation.

### The sparse quantized hopfield network

We develop an implementation of a discrete graphical model that uses primarily neural network operations. We call it the sparse quantized Hopfield network (SQHN). SQHNs have architectural similarities to Hopfield networks and Bayesian networks. Unlike standard Bayesian networks, SQHNs are relatively easy to scale and more compatible with the hardware that assumes vector-matrix multiplication as the basic operation (e.g., GPUs and memristors). Unlike common Hopfield nets, SQHNs explicitly utilize quantization and implement a discrete, directed graphical model. Hidden nodes in SQHN models are assigned integer values during inference, represented by sparse one-hot vectors. Sparsity distinguishes SQHNs from prior quantized Hopfield networks (e.g., refs. [26,27]), which assign an integer value to each neuron. The sparse code we use ensures subsets of parameters are isolated during training.

SQHNs implement directed graphical models. In this paper, we consider tree-architectures without loops (see Fig. 1), though many other architectures are possible. Visible nodes are nodes clamped to portions of the input (e.g., image patches), which are assumed, though not required, to be continuous. Each hidden node $l$ represents a categorical variable that takes an integer value represented by a one-hot vector, $h^*_l$. We also notate clamped values at visible nodes as $h^*_l$. Conditional probability $p(h^*_l|pa_l)$ of node $l$ given its parent value is parameterized by synaptic weight matrices. For example, in the simple case where $l$ has one parent $p(h^*_l|pa_l)$ it is parameterized by matrix $M_{pa_l,l}$. The energy is a summation over conditional probabilities:

$$E(h^*, \theta) = \frac{1}{L} \sum_{l=0}^{L} p(h^*_l|pa_l). \qquad (1)$$

Typically, in directed graphical models, like Bayesian Networks, the aim of MAP inference is to update the values of hidden nodes to maximize the joint probability of node states, which is the product of the conditional probabilities rather than the sum[28]. We show that the summation of the conditional probabilities approximates a kind of joint probability that takes into account uncertainty over parameters (Supplementary 1.1). Taking into account this uncertainty is crucial for learning in online settings. Further, by approximating this joint distribution with the energy above, we can implement a model that performs inference using only standard artificial neural network operations that sum inputs to neurons rather than multiply, which would be needed with the standard joint distribution (Supplementary 1.2).

**Algorithm 1.** SQHN recall algorithm

```
begin
    // Clamp visible nodes to x
    for l = 0 to L do
        // Compute FF value for h_l, equation 7, 9 in supplementary
           material
    end
    for reversed(l = 0 to L − 1) do
        // Combine h_l and FB, equation 13 in supplementary material
        // Set h*_l = argmax(h_l)
    end
    // Set each input patch, x_{c_l}, to FB given parent value h*_l
end
```

**Algorithm 2**. SQHN learning algorithm

```
begin
    // Clamp visible nodes to x
    for l = 0 to L do
        // Compute FF input h_l, equation 7, 9 in supplementary
           materials
        // If h_l < ε, set h_l* to new value.
        // Else h_l* = argmax(h_l)
    end
    // Update ε, equation 5
    // Update Weights, equation 2.2
end
```

SQHN models update neuron activities during recall in a way akin to MAP inference, where neurons are updated to maximize the energy:

$$h^* = \text{argmax}_{h^*} \cdot E(h^*, \theta, x), \tag{2}$$

where $h^*$ is the set of one-hot vectors assigned to each node. We use an inference procedure, to approximate this minimization problem. The procedure resembles the max-product algorithm[29]. Like the max-product algorithm, the SQHN inference procedure is computationally cheap, requiring only a single feed-forward/bottom-up (FF) and feedback/top-down (FB) sweep through the network. Unlike the max-product algorithm, however, the SQHN inference is more straightforward to implement, uses only standard neural network operations, is more compatible with the hardware that assumes vector-matrix multiplies (e.g., GPUs), and minimizes a different quantity than the standard max-product algorithm (see

supplementary 1.1). Mathematically, the SQHN recall process can be understood as follows: each data point $x^t$ observed during training is stored at a maxima of the energy. Therefore, given a corrupted input $\tilde{x}^t$, an SQHN can reconstruct the original by finding the latent code that maximizes the energy given $\tilde{x}^t$, which will be the same latent code as the original data point and will therefore output a close reconstruction of the original data point (Fig. 1B), where typically the higher the energy of the latent code, the better the reconstruction. Mechanistically, recall works by propagating signals up to the memory node which is assigned a memory value. Then a signal is propagated down, encouraging lower-level hidden nodes to take values associated with the memory node's value (Fig. 1C). For a pseudo-code description, see algorithm 1.

SQHNs update their parameters using an algorithm akin to MAP learning, where each training iteration the model first performs inference to maximize energy w.r.t. activities and then updates weights to further increase energy.

$$\theta^T = \text{argmax}_\theta \sum_{t=0}^{T} E(\theta, h^{*,t}, x^t), \tag{3}$$

where matrices at hidden layers must meet certain normalization constraints, making this a constrained optimization problem. Importantly, weights are updated to maximize energy over all previously observed data points rather than just the one present at the current iteration. This helps prevent forgetting of previous data points in online scenarios. However, the update is performed using only the activities and data points from the current iteration (i.e., there is no

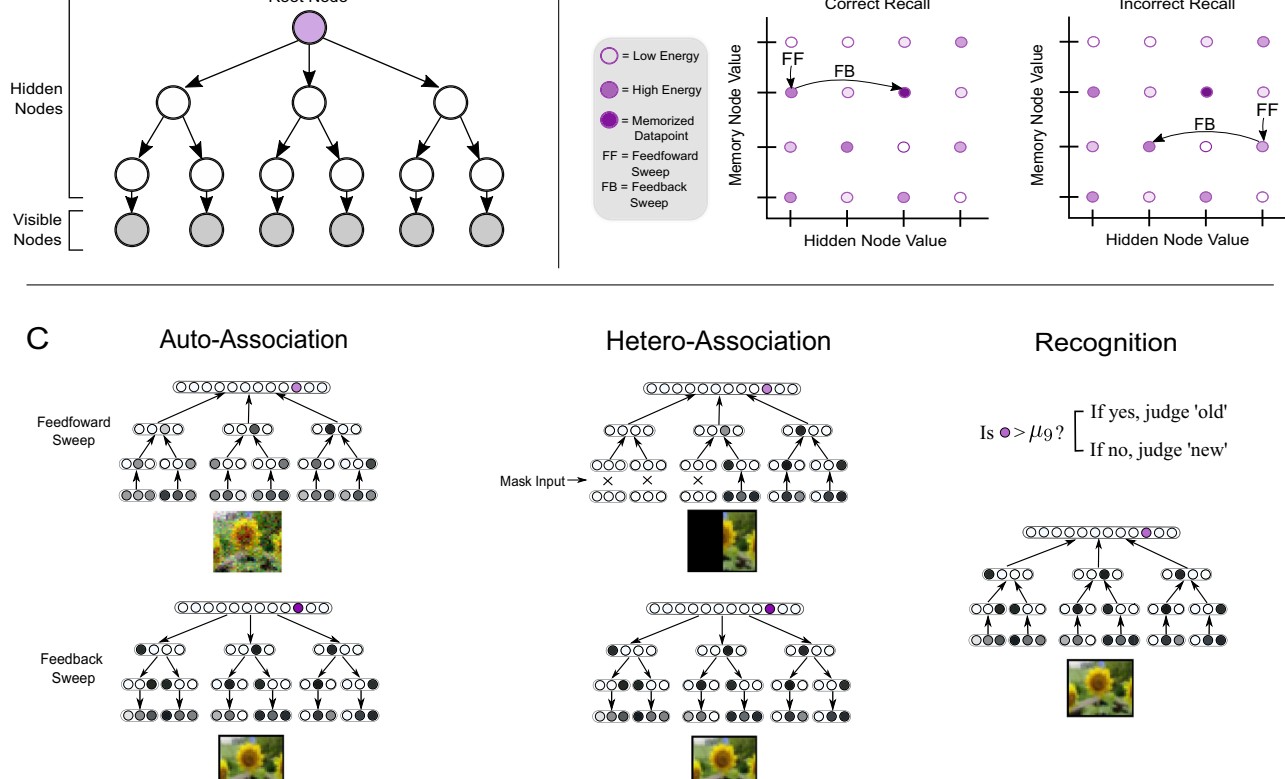

**Fig. 1 | Illustration of SQHN. A** Tree-structured, directed, acyclic graph. **B** Diagram of recall process in terms of the node values and energy. Nodes take integer values. Correct recall finds the set of node values at the global maximum of the energy. Correct recall typically only occurs if the memory node has the correct value, since the memory node is fixed after the feedforward (FF) pass, after which it then adjusts the values of hidden nodes through feedback/top-down signals. **C** Neural network

diagrams of the SQHN during associative recall and recognition. During recall, the FF sweep propagates signals up the hierarchy, where the memory/root node retrieves the most probable (high energy) value, and propagates the signals down, encouraging hidden nodes to take the values associated with that particular memory/value.

buffer of previous data points or activities). The solution is a local Hebbian-like update rule:

$$\Delta M_{pa_l,l} = \frac{1}{c^*_{pa_l}+1}(h^*_l - M_{pa_l,l}h^*_{pa_l})h^{*\top}_{pa_l},\qquad(4)$$

where $c^*_{pa_l}$ is a count of the number of iterations the parent node value was activated during training (see supplementary 1.4). Because $h^*_{pa_l}$ is a sparse, one-hot vector, this is a sparse weight update that only alters the values in a single column of matrix $M_{pa_l,l}$.

Importantly, instead of randomly initializing weights, we initialize all weights to 0, then grow new neurons and synapses as needed. In particular, a new neuron is grown at node $l$ at training iteration $t$, if no neuron at the node has a value greater than some threshold. We use an exponentially decaying threshold based on the Dirichlet prior (see supplementary 1.3):

$$\epsilon = \frac{\alpha}{(t+\alpha)},\qquad(5)$$

where $t$ is current training iteration and $\alpha$ is a hyperparameter. Ablations show that if neuron growth, decaying growth threshold, and/or learning rate decay are removed, the model performs noticeably worse in online-continual settings (Supplementary Fig. 2). For pseudo-code of the learning algorithm, see algorithm 2.

### Related works
SQHNs have similarities to recursive cortical networks (RCN)[30], another neural network/Bayesian network hybrid that uses an algorithm akin to MAP learning in a tree-structured architecture. SQHNs use different learning rules than RCNs. RCNs, for example, do not perform the averaging operation that SQHNs do, and as a result, RCNs cannot learn under noise (e.g., see supplemental from ref. [30]). RCNs also have more complex architectures that utilize max-pools and factorize color and shape representations. RCNs also do not explicitly minimize an energy function. Further, as far as we know RCNs have not been tested on natural images, or on associative memory tasks from recent machine learning literature. Other models like sparse Hopfield-like networks of refs. [31,32] and Hierarchical Temporal Memory[33], also utilize sparse neural networks with some architectural similarities to SQHNs. However, these models do not explicitly encode discrete random variables, they use distinct energy functions or no energy function at all, and they have not been applied to memory tasks we are interested in here.

Following previous reviews[12–17,34], continual and online-continual learning approaches may be split into several types: (1) Regularization-based approaches constrain the way parameters are updated to reduce catastrophic forgetting (e.g., refs. [35–39]). (2) Memory-based approaches store or model previously observed data for the purpose of replaying the data during training (e.g., refs. [40–42]). (3) Parameter isolation models avoid forgetting by allocating different parameters for each task, either by gating components or by dynamically adding new sub-networks as needed (e.g., refs. [22–25]). Our SQHN model differs from these approaches since it uses local learning rules, while these previous approaches use BP in some form (e.g., all the methods reviewed by refs. [14,16] use BP either in pre-training and/or during online training of the classifier). Our SQHN model uses a kind of parameter isolation (via sparse neuron activity and neuron growth) combined with regularization (via learn rate decay schedules) to avoid forgetting. As far as we can tell, SQHN's particular strategy is distinct from these approaches. For example, whereas all the previous works on regularization and parameter isolation use some form of global gradient information to regularize and isolate parameters, the SQHN does so based solely on information local to the neuron or synapse.

Like the SQHN, classic Hopfield networks[43] and modern Hopfield networks (MHN)[44–46] are energy-based models that can perform auto-associative memory tasks. Of existing Hopfield nets, the SQHN is the most closely related to the continuous modern Hopfield network (MHN)[45]. The continuous MHN performs recall using the operation

$$x_{\text{new}} = M\,\text{softmax}(\beta M^T x),\qquad(6)$$

where $M$ is a matrix of data points/memory vectors, $x$ is a matrix of input/query vectors, and $\beta$ is the inverse temperature. It was shown by ref. [45] that this model is a special case of the attention layer in the transformer. One interpretation of this operation is that it is performing a kind of nearest neighbor computation (e.g., ref. [46]), where similarity values between memory vectors and query vectors are computed using the dot product. A weighted average of memory vectors is returned, where those memory vectors more similar to the query vector are given more weight. A simple SQHN network with a single hidden layer can be interpreted as performing a kind of nearest neighbor operation as well, where $\beta = \infty$:

$$x_{\text{new}} = M\,\text{argmax}\left(\frac{1}{Z}(M^T - \mu)(x - \mu)\right),\qquad(7)$$

where $\mu$ is the scalar that shifts the matrix M (we set $\mu = 0.5$ below, see methods), $\frac{1}{2}$ normalizes each elements of the hidden layer input such that the values range between zero and one, and the input, x, and shifted memory vectors, $m_j - \mu$, are normalized.

Despite both models using a kind of nearest neighbor operation, SQHNs differ from MHNs in ways that provide the SQHN significant advantages in the tests below. First, when the memory matrix, $M$, is trainable (e.g., as in ref. [44]), the MHN uses a randomly initialized fixed size $M$, which is trained with BP. The SQHN, on the other hand, uses a non-random $M$, which dynamically adds memory vectors as needed, and uses a local energy-based rule to update existing memory vectors specifically to solve the online learning problem noted above. Below, we find the MHN learns slowly and suffers from significant catastrophic forgetting, whereas the SQHN learns very fast and shows minimal forgetting. Second, we found that the explicit normalization of the input and the hard, winner-take-all recall performed by the SQHN, was superior to operations of the MHN which do not explicitly normalize input and use softmax. Though softmax approaches the argmax as $\beta \to \infty$, very large beta values yielded worse performance in scenarios where the matrix $M$ had to be trained, forcing the MHN to perform a kind of soft recall in these scenarios making accurate recall difficult.

SQHNs also have interesting similarities to predictive coding networks (PCNs)[47], which have been used recently for associative memory tasks (e.g., see refs. [19,48–50]). Both SQHNs and (PCNs) can be interpreted as directed graphical models and utilize a MAP learning algorithm[46]. The central difference between the two is that SQHNs have categorical random variables at hidden nodes, while PC networks have Gaussian random variables. As such, SQHNs and PC networks minimize distinct energies and use distinct inference procedures. PCNs minimize the free energy (F):

$$F = \sum_{l=0}^{L}\frac{1}{2}\parallel h_l - W_{pa_l,l}h_{pa_l}\parallel^2,\qquad(8)$$

where $h_l$ is the vector at node $l$ representing the value of the Gaussian mean at that node, and $h_{pa_l}$ is the mean vector at the parent node of $l$. The matrix $W_{pa_l}$ parameterizes the conditional probability distribution over $l$ given the parent node value. Interestingly, PCNs and SQHNs have very similar learning rules. The PCN learning rule, applied at the end of the inference phase, is the gradient of the free energy

$$\Delta W_l = -\alpha\frac{\partial F}{\partial W_l} - \alpha(h_l - W_{pa_l}h_{pa_l})h^{\top}_{pa_l},\qquad(9)$$

which we can see is identical to the SQHN rule up to a scalar learning rate $\alpha$.

Despite these similarities, SQHNs show advantages over PCNs in several respects. First, the iterative, gradient-based inference procedure of PCNs is computationally costly, requiring dozens, sometimes hundreds, of updates to neurons each iteration to achieve good performance on associative memory tasks (e.g., see refs. [19,48]). This effectively renders PCNs impractical for online learning, which favors fast, computationally cheap models. SQHNs, on the other hand, perform only a single FF and FB sweep through the network to perform inference. Secondly, we hypothesized above that continuous-valued latent variables are necessarily more sensitive to corrupted inputs, since such variables may take an infinite number of values, and, therefore can always adjust hidden values to increase the likelihood of corruption in the data. MAP inference in discrete models, on the other hand, can better ignore corruption since its latent code only has a finite number of values, and can therefore map similar versions of the same input to the same latent code, yielding exact reconstruction. This hypothesis is supported in our experiments, where we find SQHNs are much more robust to noisy inputs than PCNs.

## Experiment: auto-association and hetero-association comparison

We tested SQHN on several tasks and compared its performance to SoTA and baselines. Specifically, we tested performance on: (1) Auto-association. (2) Hetero-association. (3) Online-continual auto-association. (4) Noisy encoding. (5) Episodic memory.

We first compared SQHN models to SoTA associative memory models on auto-associative and hetero-associative recall tasks. For both tasks, unaltered data points from a set $X_{train}$ are presented to the model during training. In auto-association, during testing, the model is given corrupted versions, $\tilde{X}_{train}$, of training data, and the model is tasked with reconstructing the original data points. Here, corruption is added to the images with white noise. For hetero-association tasks, during testing portions of the input data are treated as missing. Following, the recent work of ref. [19], we remove a certain number of pixels from the input data randomly (pixel dropout), or we remove a certain number of the rightmost pixels (mask).

Three SQHNs are tested: an SQHN with one hidden layer (SQHN L1), two hidden layers (SQHN L2), and three hidden layers (SQHN L3). We compare to two types of SoTA models: predictive coding networks (PCN) and modern Hopfield networks (MHNs). PCNs are neural network models[51], that implement a kind of probabilistic generative model with continuous latent variables[47]. Three types are compared: offline-trained PCN (GPCN)[48] and two online-trained versions (BayesPCN, BayesPCN with forgetting)[19]. Continuous MHNs[45] have similarities to auto-encoders with a single hidden layer where a softmax activation is used. We compare it to the original model of ref. [45] (MHN), a version of this model by ref. [52], which showed better performance by using a Manhattan distance measurement in its recall operation (MHN-Manhtn) (see methods), and the MHN of ref. [19] that used a gradient-based inference procedure (MHN-GradInf).

Results are in Table 1. PCN models struggled with noisy, auto-association task. The MHN that uses Manhattan distance performed very well, and the MHN-GradInf and the one-level SQHN model performed perfectly on all auto-association tests. The multi-level SQHNs were more sensitive to white noise on smaller CIFAR-10 images, but still performed well with moderate corruption, and performed very well when they have larger lower-layer receptive field sizes, which they use on the larger Tiny Imagenet images.

The GPCN and BayesPCN achieved very low recall MSE on most masking tasks. MHN-grad and the MHN with Manhattan distance performed well on moderate masking, but failed completely on the high masking scenario. The one and three level SQHN models performed perfectly on all masking tasks, while the two level performed nearly

perfectly. SQHN models were the only models to match SoTA performance across both auto-associative and hetero-associative tasks.

## Experiment: online, continual auto-association

Next, we test SQHNs on online, continual auto-association. The application for online learning algorithms are typically embedded learning systems (e.g., robots, sensing devices, etc.), where computationally and memory-efficient algorithms are preferred. PCN networks are highly computationally expensive, requiring hundreds of neuron updates per training iteration (see refs. [19,48]), making them impractical as an online auto-associative memory system. Thus, we compare SQHNs to the computationally efficient MHN model. However, we cannot perform the batch update that is typically used in auto-associative memory tests of MHNs. Instead, following previous work (e.g., refs. [44,45]), we train MHNs with BP to reduce reconstruction error. As baseline comparisons, we train MHNs with BP/SGD and BP with an Adam optimizer. We also train with several compute-efficient algorithms common to continual learning: online elastic weight consolidation (EWC++)[53], which is a kind of regularized SGD, and episodic recall (ER)[40], which uses a small buffer to store a mini-batch of previously observed data points and SGD to update weights with the mini-batch.

We test on two kinds of online-continual auto-associative tasks: online class incremental (OCI) and online domain incremental (ODI). In both, data from each task is presented incrementally, one task at a time. In the OCI setting, each task consists of images from the same class. In the ODI setting, there are four data sets, each composed of visually distinct images (e.g., dataset 1 has bright images, dataset 2 has dim images, etc.). During training, models perform a single pass over each dataset, observing only a single data point at each iteration, before switching to the next dataset. At testing, a noisy version of previously observed data were presented.

Performance is measured using recall MSE ($\mathcal{L}_{MSE}$) and following previous works[19,48] recall accuracy:

$$\mathcal{A}^T = \frac{1}{T}\sum_{t=0}^{T}\mathbf{1}\left(\frac{1}{d}\|x^t - x^{t,new}\|^2 < \gamma\right),\qquad(10)$$

where the indicator function $\mathbf{1}$ is one if the recall MSE is below threshold, $\gamma$, and zero otherwise. We also use a 'cumulative' (i.e., average) performance measure, which is common in online learning scenarios:

$$\mathcal{C}_{MSE} = \frac{1}{T}\sum_{t=0}^{T}\mathcal{L}_{MSE}^t, \quad \mathcal{C}_{Acc} = \frac{1}{T}\sum_{t=0}^{T}\mathcal{A}^t,\qquad(11)$$

where $\mathcal{L}_{MSE}^t$ and $\mathcal{A}^t$ is the recall MSE and recall accuracy, respectively, at iteration $t$ given the model parameters and input data at iteration $t$. Cumulative measures are sensitive not just the final performance, but also to sample efficiency, i.e., how quickly the models improves performance. Finally, design and use a measure of sensitivity to data ordering ($S_{MSE}$):

$$S_{MSE} = \left|\mathcal{C}_{MSE}^{OnCont} - \mathcal{C}_{MSE}^{On}\right|.\qquad(12)$$

This sensitivity measure is 0 when the model achieves the same cumulative MSE in the online (On) and online-continual (OnCont) settings, and increases as the performance differs. Models insensitive to ordering are highly useful in realistic scenarios where the way data is presented to the model is difficult to control and predict.

Results are shown in Fig. 2. SQHNs were highly insensitive to the ordering of the data in online-continual scenarios (Fig. 2C, top and bottom). They perform one-shot memorization until their capacity is reached, and then their performance decays slowly (Fig. 2A, top and bottom), yielding very good cumulative performance (Fig. 2B, top and

**Table 1 | Top**

| Recall MSE - Moderate corruption | | | | | | |
|---|---|---|---|---|---|---|
| | White noise | | Pixel dropout | | Mask | |
| | CIFAR-10 | TinyImgNet | CIFAR-10 | TinyImgNet | CIFAR-10 | TinyImgNet |
| GPCN(offline)[19] | $0.0121^{(\pm0.0001)}$ | $0.0067^{(\pm0.0004)}$ | $0.0001^{(\pm0.0000)}$ | $0.0000^{(\pm0.0000)}$ | $0.0009^{(\pm0.0000)}$ | $0.0001^{(\pm0.0000)}$ |
| BayesPCN[19] | $0.0337^{(\pm0.0007)}$ | $0.6606^{(\pm0.0267)}$ | $0.0001^{(\pm0.0000)}$ | $0.0000^{(\pm0.0000)}$ | $0.0019^{(\pm0.0000)}$ | $0.0000^{(\pm0.0000)}$ |
| BayesPCN(forget)[19] | $0.0188^{(\pm0.0002)}$ | $0.0176^{(\pm0.0001)}$ | $0.0019^{(\pm0.0000)}$ | $0.0008^{(\pm0.0000)}$ | $0.0465^{(\pm0.0001)}$ | $0.0235^{(\pm0.0001)}$ |
| MHN | $0.1457^{(\pm0.0097)}$ | $0.0955^{(\pm0.0073)}$ | $0.4512^{(\pm0.0558)}$ | $0.4545^{(\pm0.0456)}$ | $0.5538^{(\pm0.0149)}$ | $0.6101^{(\pm0.0091)}$ |
| MHN-Manhtn | $0.0000^{(\pm0.0000)}$ | $0.0000^{(\pm0.0000)}$ | $0.0000^{(\pm0.0000)}$ | $0.0000^{(\pm0.0000)}$ | $0.0020^{(\pm0.0028)}$ | $0.0016^{(\pm0.0012)}$ |
| MHN-GradInf[19] | $0.0000^{(\pm0.0000)}$ | $0.0000^{(\pm0.0000)}$ | $0.0000^{(\pm0.0000)}$ | $0.0000^{(\pm0.0000)}$ | $0.0000^{(\pm0.0000)}$ | $0.0001^{(\pm0.0000)}$ |
| **SQHN L1** | $0.0000^{(\pm0.0000)}$ | $0.0000^{(\pm0.0000)}$ | $0.0000^{(\pm0.0000)}$ | $0.0000^{(\pm0.0000)}$ | $0.0000^{(\pm0.0000)}$ | $0.0000^{(\pm0.0000)}$ |
| **SQHN L2** | $0.0002^{(\pm0.0001)}$ | $0.0000^{(\pm0.0000)}$ | $0.0000^{(\pm0.0000)}$ | $0.0002^{(\pm0.0002)}$ | $0.0001^{(\pm0.0000)}$ | $0.0008^{(\pm0.0002)}$ |
| **SQHN L3** | $0.1076^{(\pm0.0022)}$ | $0.0000^{(\pm0.0000)}$ | $0.0000^{(\pm0.0000)}$ | $0.0000^{(\pm0.0000)}$ | $0.0000^{(\pm0.0000)}$ | $0.0000^{(\pm0.0000)}$ |
| Recall MSE - High Corruption | | | | | | |
| | White Noise | | Pixel Dropout | | Mask | |
| | CIFAR-10 | TinyImgNet | CIFAR-10 | TinyImgNet | CIFAR-10 | TinyImgNet |
| BayesPCN[19] | $0.0755^{(\pm0.0002)}$ | $0.0242^{(\pm0.0002)}$ | $0.0000^{(\pm0.0000)}$ | $0.0000^{(\pm0.0000)}$ | $0.0006^{(\pm0.0000)}$ | $0.0001^{(\pm0.0000)}$ |
| MHN-Manhtn | $0.0000^{(\pm0.0000)}$ | $0.0000^{(\pm0.0000)}$ | $0.1840^{(\pm0.0368)}$ | $0.1248^{(\pm0.0884)}$ | $0.7644^{(\pm0.2844)}$ | $0.7588^{(\pm0.2188)}$ |
| MHN-GradInf[19] | $0.0052^{(\pm0.0000)}$ | $0.0000^{(\pm0.0000)}$ | $0.3840^{(\pm0.0010)}$ | $0.5630^{(\pm0.0036)}$ | $0.3957^{(\pm0.0000)}$ | $0.6378^{(\pm0.0000)}$ |
| **SQHN L1** | $0.0000^{(\pm0.0000)}$ | $0.0000^{(\pm0.0000)}$ | $0.0000^{(\pm0.0000)}$ | $0.0000^{(\pm0.0000)}$ | $0.0000^{(\pm0.0000)}$ | $0.0000^{(\pm0.0000)}$ |
| **SQHN L2** | $0.0904^{(\pm0.0078)}$ | $0.0002^{(\pm0.0004)}$ | $0.0000^{(\pm0.0000)}$ | $0.0000^{(\pm0.0000)}$ | $0.0000^{(\pm0.0000)}$ | $0.0008^{(\pm0.0006)}$ |
| **SQHN L3** | $0.3324^{(\pm0.0056)}$ | $0.0240^{(\pm0.0032)}$ | $0.0000^{(\pm0.0000)}$ | $0.0000^{(\pm0.0000)}$ | $0.0000^{(\pm0.0000)}$ | $0.0000^{(\pm0.0000)}$ |

Recall MSE of 1024 images from CIFAR-10 and Tiny ImageNet data sets. At test time, images are either corrupted with white noise (variance 0.2), 25% of the pixels dropped, or the right 25% of the pixels masked. Bottom Recall MSE of 128 images from CIFAR-10 and Tiny ImageNet data sets. At test time, images are either corrupted with white noise (variance 0.8), 75% of the pixels dropped, or the right 75% of the pixels masked.

bottom). The BP-based MHNs learned too slowly to recall any data points and were much more sensitive to ordering, especially in the ODI setting. This leads to poor cumulative scores. These results provide evidence the SQHN is a highly effective online-continual learner, especially in scenarios where fast learning is essential, and may have general benefits over SGD/BP-based approaches.

### Experiment: noisy encoding
We designed and tested SQHN models on a noisy encoding task. In this task, in each training iteration, some number of noisy samples of an image are generated and presented to the network one at a time. Images are Gaussian (white noise added) or binary samples. The models must encode the images, and reconstruct them at test time, when they are presented with the original non-corrupted versions. At no point during training is the model shown the non-corrupted version of the data. The purpose of this task is to specifically test the model's ability to demonstrate unsupervised learning in a noisy setting.

We compared the one and three hidden layer SQHN and MHN models on E-MNIST and CIFAR-100, respectively. The one-layer models had 300 hidden layer neurons, while the three hidden layer architectures had 150 neurons at each node. Networks were presented with 300 and 150 noisy images, respectively, so an inability to recall the images was more dependent on an inability to remove noise during learning rather than on capacity constraints.

In addition to testing the SQHN and MHN models, we tested an SQHN model with a slight alteration (SQHN+), where after the hidden states are computed for the first image sample, the hidden states are held fixed for the remaining duration of the training iteration. This ensures the latent code did not change during the encoding of the same image.

SQHN models, especially SQHN+, were highly effective at removing noise during learning (Fig. 3), and both SQHN models significantly outperformed MHNs trained with BP and BP-Adam. This

provides evidence the SQHN can be an effective learner under noise and outcompete similar BP-based models.

### Experiment: episodic memory
Next, we designed and tested models on an episodic memory task inspired by human and animal memory experiments, which we call episodic recognition. Recognition is a hallmark ability of animal memory but has not, as far as we can find, been developed into a formalized machine-learning test. In our task, models are presented with a sequence of training data points, $X_{train}$, in an online fashion. Afterward, the model is tested on a binary classification task, where the model must classify the data point presented as old (in the training set) or new (not from the training set). The testing dataset is an equal portion of observed training images, $X_{train}$, new unobserved images from a related (in distribution) data set, $X_{new-in}$, and new images from an unrelated (out of distribution) data set, $X_{new-out}$. A high-performing memory system will achieve significantly above-chance accuracy, which will decay gracefully as the size of the data set increases.

We develop a principled method for the SQHN to perform recognition. This method uses the value of the neuron at the root/memory node with the maximum value. This maximum value tells us how similar the features of the current data point is to the features of the most similar previously observed data point (Supplementary Note 5). If the max activity at the memory node is above some threshold, the data point is judged to be old. If below, it is judged to be new. The threshold we use is the moving average, $\mu_l$, of the activity values observed for each neuron:

$$\mu_{L,j}^t = \frac{c_{L,j}^t - 1}{c_{L,j}^t} \mu_{L,j}^{t-1} + \frac{1}{c_{L,j}^t} h_{L,j}^t, \qquad (13)$$

where $j$ is the neuron with the maximum value, $h_{L,j}$, at root node $L$ at iteration $t$. We show that this value is an estimate of the desired

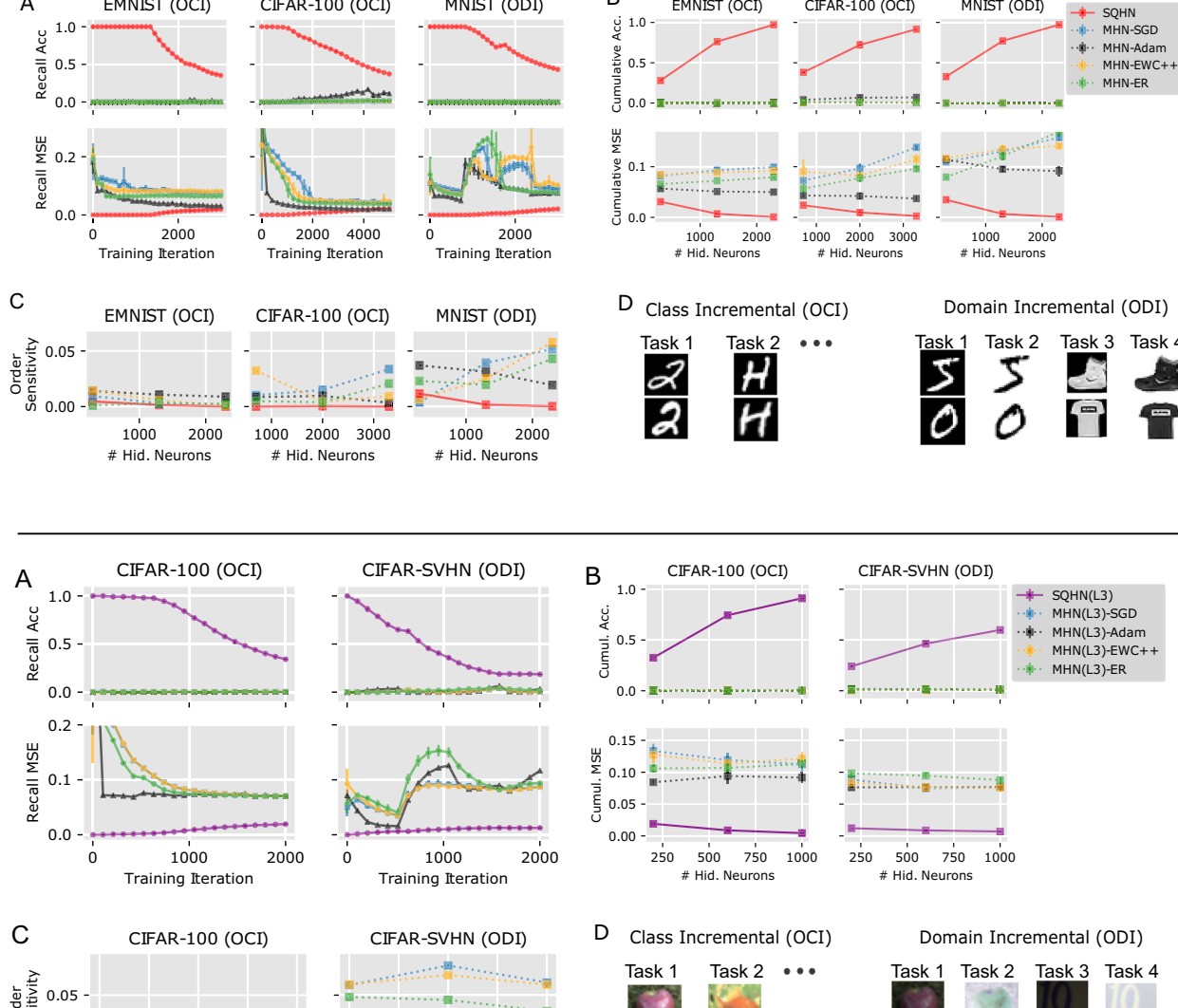

**Fig. 2 | Online-continual auto-association.** Top: One hidden layer models with small (300), medium (1300), and large (2300) node sizes. Bottom: Three hidden layer models with small (200), medium (600), and large (1000) node sizes. **A** Recall accuracy and recall MSE during training on models with medium-sized hidden nodes. **B** Cumulative recall MSE and recall accuracy for each model size. **C** Order sensitivity for each model size. **D** Online class incremental (OCI) versus online domain incremental (ODI) settings.

probability of $p(x^t = old|\theta^{(t-1)}, x^t) = 0.5$, which is the threshold at which it becomes more probable than not that the data point is old rather than new (Supplementary Note 5). For a pseudo-code description of recognition, see Supplementary Algorithm 3.

Since there are no previous baselines to compare SQHN against for the episodic recognition task that we know of, we ran a simple comparison between an SQHN and MHN with single hidden layers (Fig. 4). Since MHN has not been used for an episodic recognition task, we created two methods for performing recognition in an MHN with one hidden layer. The first method uses the activities at the hidden layer as a measure of similarity to stored data points. The second method keeps a moving average of the recall MSE during training. If the hidden layer activity is above or if the MSE is below a threshold, the model judges the data point as new. A grid search is used to set the threshold parameter.

SQHN models performed perfectly until capacity is reached (vertical dotted line). The performance then decayed gradually,

whereas MHN models were unable to do better than chance (Fig. 4A). The performance differences seem due to the fact that SQHN models 'overfit' the training data (Fig. 4B), early in training, allowing it to recognize $X_{new-in}$ as less probable than $X_{old-in}$. The MHN model, on the other hand, performed identically with old and new in-distribution data. This allowed MHN to generalize better earlier, but it prevented the MHN from being able to distinguish previously observed data points from similar data points that were not present in the training distribution.

**Experiment: further comparison of SQHN architectures**
Finally, we did a thorough comparison of SQHN models that have one, two, or three hidden layers (Fig. 5A), to better establish what the advantages and disadvantages are of adding more hidden layers to the SQHN.

First, we tested how sensitive SQHN models are to corruption in several auto-association tasks (Fig. 5B). During training each model

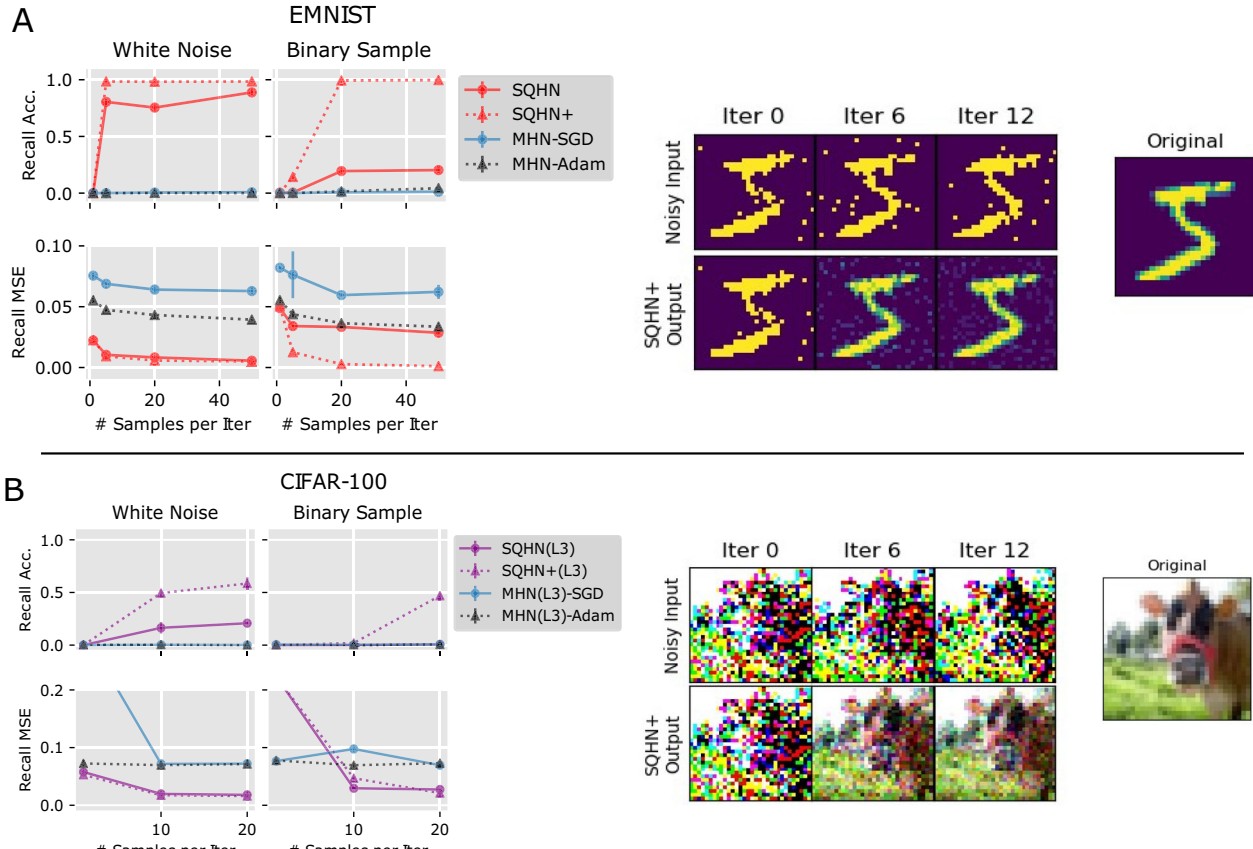

**Fig. 3 | Noisy encoding task. A** Recall accuracy and recall MSE for one hidden layer models under white and noise and binary sample conditions. Example of the reconstruction during test time for SQHN+ model, in the case of 1, 6, and 12 samples. **B** Recall accuracy and recall MSE for three hidden layer models under white and noise and binary sample conditions with reconstruction example for SQHN on the right.

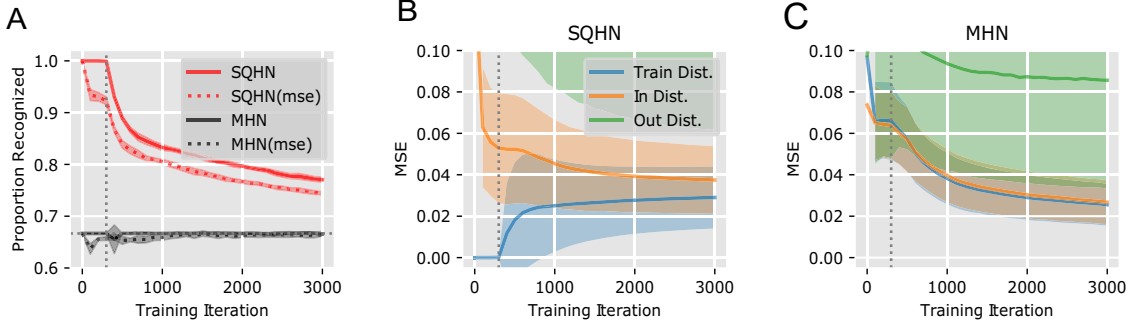

**Fig. 4 | Recognition task. A** The recognition accuracy for SQHN and MHN are shown in networks with 300 neurons at the hidden layer (300th iteration marked by a vertical dotted line). **B** The MSE for the SQHN model on the training MNIST data (Train Dist.), hold-out MNIST data (In Dist.), and the F-MNIST data (Out Dist.) **C** The MSEs for the MHN model.

memorizes 1000 images from the CIFAR-100, TinyImageNet, or CalTech 256 data sets. During testing, images are either corrupted with white noise or, what we call, an occlusion. In the occlusion scenario, pixels in a rectangular region of random shape and position are set equal to either 0 (black occlusion), a random color (color occlusion), or white noise (noise occlusion). (Note this is distinct from masking, since models treat pixels as corrupted rather than missing.) Results are in Fig. 5B. All SQHN models performed recall perfectly when corruption was small. Adding more hidden layers tended to improve performance on occlusion tasks, likely because trees represent data as a part-whole hierarchy and, therefore, can better ignore corrupted parts of the input during inference. Architectures whose bottom hidden

layer nodes had larger receptive fields performed better on the noise task. The one-layer SQHN had the largest receptive field, so it performed the best. For the largest images (CalTech256), however, receptive field sizes at the bottom layers were large for all models (minimum $8 \times 8$), and all models performed similarly on noisy recall.

Next, we observed auto-associative recall performance during online (i.i.d.) learning for SQHN models with different numbers of layers and node sizes (Fig. 5D). All SQHN models performed one-shot memorization until minimum capacity is reached (which is the number of neurons at hidden nodes, see theorem 1 Supplementary Note 6). Deeper SQHN's recall accuracy decayed at a much slower rate. We suspected this was due to their ability to reuse primitive feature

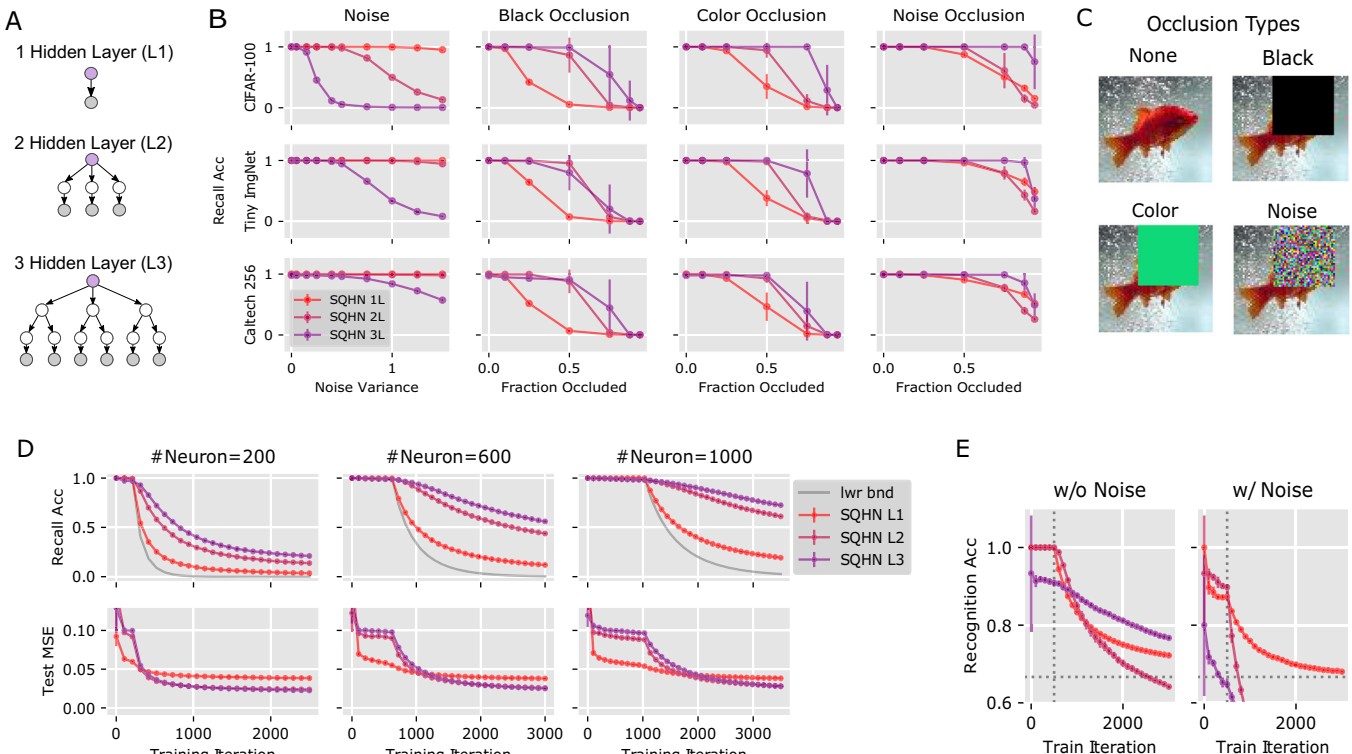

**Fig. 5 | Comparison of SQHN architectures with different numbers of hidden layers. A** Depiction of various SQHN architectures. **B** Recall accuracy across three data sets (CIFAR-100, Tiny Imagenet, Caltech 256) under the white noise and several occlusion scenarios. **C** Visualization of the black, color, and noise occlusions. **D** Recall accuracy during online training without noise (top row) and MSE on a test a test set (bottom row). The lower bound on recall accuracy posited by theorem 2 (supplementary note 6) is marked by a gray line. Models tested with different maximum number of neurons per node (200, 600, 1000). **E** Recognition accuracy for three SQHN models, with and without noise. 500 neurons are allocated to each node (vertical dotted line marks when all 500 neurons are grown). CIFAR-100 data was used for train and in-distribution set, while a flipped pixel version of the Street View House Numbers (SVHN) dataset was used for out-of-distribution. The best guessing strategy yields 66% accuracy (horizontal dotted line).

representations at lower nodes. It is well known natural images are well described as compositional hierarchies of a small set of primitive features. By learning a small set of primitive features at lower layers, multi-level SQHNs seem able to generalize these learned features across more data points, increasing capacity and slowing the forget rate. We tested these models on test/hold-out data and indeed found the multi-level tree-architectures generalized better to new data than the one hidden layer model.

Finally, we compared SQHN models on the episodic recognition task. (Fig. 5E). Models had 500 neurons at each node. The $X_{train}$ data were CIFAR-10 train images, $X_{new-in}$ were CIFAR-10 images from a hold/out test set, and $X_{new-out}$ were images from the SVHN dataset with flipped pixels. Figure 5E shows all SQHN models were able to perform well above-chance accuracy (66%, horizontal dotted line) even when far more training data was presented than the number of neurons at the memory node.

In sum, adding more levels to tree-structured SQHNs significantly improves auto-associative recall with occlusion, slows the decay of recall accuracy after the network hits capacity, and improves generalization without significant loss in recognition ability. While adding levels can make the SQHN more sensitive to noise, this seems limited to small images.

## Discussion

Artificial neural networks were originally designed to mimic the way biological neural circuits process information[54]. The way biological circuits learn to process information, however, is an open question. In particular, it is unknown how the brain uses local learning rules effectively in noisy, online-continual settings. In this paper, we proposed a general local learning approach to the basic task of storing and retrieving patterns in noisy, online-continual scenarios. We proposed using a sparse, quantized neural code to deal with noisy and partial inputs and to prevent catastrophic forgetting, and implementing this strategy via a discrete graphical model that performed MAP learning, an algorithm that uses local learning rules. We implemented this approach in the SQHN model.

Our results support the effectiveness of our approach and the SQHN. First, we found the sparse quantized neural code of the SQHN was advantageous in auto-associative recall over similar models that use a dense, continuous latent code. PC models, for example, like the SQHN, implement directed graphical models and learn via MAP learning[46]. However, because PC models use a continuous latent code, they were much more sensitive to noise than SQHN models. This lends credence to the idea that using a quantized neural code helps significantly with auto-association. MHN models performed similarly to SQHNs on the noise auto-association task. This is likely because, under the hyperparameter settings that yielded the best performance, these MHNs essentially implemented a discrete latent code (see section "Related works"). However, the operation MHNs use to map inputs to latent codes, is not as effective as that of SQHNs in the high masking settings.

Second, we found SQHNs significantly outperformed similar MHN architectures trained with BP-based algorithms on online-continual learning tasks. The SQHN trained faster, demonstrated one-shot memorization, showed only a small, stable forgetting rate, and was largely insensitive to ordering. It achieved all of this while using little extra memory, no episodic memory buffer, and little compute. Part of these performance advantages may be attributable to the parameter

isolation approach generally. However, the SQHN also uses an effective learning rate schedule to prevent forgetting (Fig. 2), and we also find mathematically that using MAP inference to set the one-hot values at hidden nodes is a principled way of deciding which parameters to update. In particular, it yields a set of activities and updates that require only a small change to existing parameters (Supplementary Note 6). This suggests the exciting prospect that MAP learning in discrete models, like SQHNs, provides a justified and bio-plausible way to perform parameter isolation, which unlike previous parameter isolation methods (e.g., refs. 22–25), does not require the computation of non-local global loss gradients to decide how to isolate parameters.

Third, the sparse quantized code reduced the negative effects of noise during the noisy encoding process by yielding a hidden latent code that was largely stable across noisy samples. If we fixed the latent code to be perfectly stable, as we did with SQHN+, then performance improves even more. The sparse updates also helped prevent noise from interfering with previously recorded memories.

Finally, the SQHN proved to be highly effective in the episodic recognition task. The memory node of the SQHN stored an explicit, itemized record of previously observed feature representations of input data (see Supplementary Note 5). During recall, the memory node performs a nearest neighbor operation by finding the item with the highest energy. This operation turned out to yield a straightforward method for detecting new versus old data points, is similar to classic cognitive models of episodic recognition (e.g., ref. 55), and it performed well even when the memory node was pushed past its capacity. Models like the MHN trained with BP, on the other hand, did not naturally learn an explicit record of previously observed feature prototypes and struggled to distinguish new from similar, old data points.

Importantly, SQHN's energy function yields a straightforward way to implement a directed graphical model with largely standard artificial neural network operations. In particular, since the energy was a sum of probabilities, rather than a product, neurons in the SQHN architecture summed inputs from children and parent nodes rather than multiplied (as belief networks do[28]), yielding standard neural network operations which are easily scaled and more compatible with hardware built specifically to handle vector-matrix multiplies. SQHN does so without the need to move probabilities to the log domain, which can sometimes yield unusual properties like the need to represent very large negative numbers (i.e., overflow issues). The energy can also be justified as an approximation to the joint distribution when uncertainty (i.e., a prior distribution) is placed over parameters (supplementary note 1). Taken together these results suggest that our general approach, and the SQHN implementation of it, could provide a highly promising basis for building neuromorphic online-continual learners.

It is also interesting to note the striking similarities between SQHNs and models of memory from neuroscience. In particular, in addition to utilizing sparse codes and local learning rules like the brain, the memory node of the SQHNs has similarities to the hippocampus (HP). HP is a region of the brain closely tied to episodic memory. We showed in our experiments the memory node, like the HP, is highly effective for episodic recognition. Further, the HP, like the memory node in SQHNs, is often proposed to be at the top of the cortical hierarchy[56], and some theories propose a central function of the HP is to retrieve and return previously stored patterns given partial, noisy inputs from cortex[57,58]. This is precisely what the memory node of SQHNs does (see Fig. 1). Maybe most interesting, we find that, on average, neurons grow more rapidly and for longer periods in the memory node than the rest of the network and synapses in the memory node tend to have larger step sizes (more flexibility) on average (Supplementary Fig. 3). This is highly consistent with observations that HP grows neurons into adulthood while the rest of the cortex stops in early adulthood[59], and the observation that synapses

are highly flexible in HP compared to the rest of the cortex[60]. Importantly, the SQHN was not engineered to have these properties. Rather these properties emerged from the SQHN learning algorithm as it solved the online-continual learning problem.

Future work will need to assess whether and how the SQHN provides possible new ideas or insights into these topics in neuroscience. Further, on the machine learning side, future work will need to assess the SQHN on tasks that require generalization, such as classification tasks. Although we found SQHNs with simple tree-architectures to be highly efficient at storing and retrieving training data, and preventing forgetting, we believe somewhat more complicated SQHN architectures will be needed for high performance on tasks like classification or self-supervised learning where generalizing to new data points is needed. Nonetheless, the results here suggest SQHNs could provide a promising approach for learning these tasks in the online-continual setting.

## Methods
### Datasets and hyperparameters
Image values ranged between 0 and 1, unless otherwise noted. Images are converted to pytorch tensors, but no normalization or other alterations were made unless otherwise specified. MNIST, Fashion-MNIST, and EMNIST are image data sets with images sized $1 \times 28 \times 28$. SVHN, CIFAR-10, and CIFAR-100 are natural image data sets with images sized $3 \times 32 \times 32$. Tiny ImageNet is a natural image data set with images sized $3 \times 64 \times 64$. Finally, CalTech256 is a natural image data set with images of various sizes. We cropped all CalTech256 images to $3 \times 128 \times 128$. For all models, we use a grid search to find hyperparameters.

Unless otherwise specified, we use a recall threshold, $\gamma$, of .01. Prior works (e.g., refs. 19,48) use smaller thresholds of 0.005 or 0.001. We use a slightly larger threshold here, since we find one of our main comparison models, MHN, is unable to recall any images while training with BP, and we wanted to show this was not simply a result of an arbitrarily small recall threshold.

### SQHN implementation details
All SQHN architectures are set up to have a tree structure. One can think of the structure as being similar to locally connected networks or convolutional networks without weight sharing. Thus, we can talk about SQHNs as having a certain number of channels at each hidden layer (equivalent to the number of neurons at each node) and the receptive field size of each channel/node. To simplify the architectures, we design SQHNs so that nodes have non-overlapping receptive fields, which means each node has one parent. More complex versions of SQHNs can be used for more complex vision tasks, but we found these simpler architectures performed very well and eased scaling in associative memory tasks. Here we explain how inference is implemented. Inference involves a single feed-forward and feedback sweep through the network. The goal of inference is to maximize energy, i.e., the sum of conditional probabilities of input and hidden node values (equation (1)).

Let $h_{l,j}$ be the internal state value of the $j$th neuron in the $l$th node. Let the $l$th node be in the first hidden layer, which has one child node. Let $M_l$ be the matrix from node $l$ to its child node, which is visible and clamped in image patch $x_{c_l}$. The matrix $M_l$ can be decomposed into columns or 'memory vectors': $M_l = [m_{l,0}, m_{l,1}, ..., m_{l,j}]$. During the feed-forward pass $h_{l,j}$ receives the signal

$$h_{l,j} = \frac{.5(m_{l,j}^\top - .5)(x_{c_l} - .5)}{\| (m_{l,j}^\top - .5) \| \| (x_{c_l} - .5) \|} + .5. \qquad (14)$$

This operation is a kind of shifted cosine similarity operation between in input and each memory vector $m_{l,j}$. We show in supplementals this operation can be interpreted as a weighted, normalized

average of the probability of each pixel value, under the assumption each pixels value is a binary variable (Supplementary Note 2). In matrix form, this operation only involves a vector-matrix multiply, like a neural network, with an extra shift operation and normalization operation:

$$h_l = \frac{1}{2Z}(M_l^\top - .5)(x_{c_l} - .5) + .5, \tag{15}$$

where $\frac{1}{Z}$ is the neuron-wise normalization. In practice, one could also store a separate feedback matrix with the shifted, transposed, and normalized version of matrix $M_l$.

Hidden nodes at the second layer and higher have children nodes that represent discrete/categorical variables. Computing the sum of probabilities of each child therefore requires a different operation. Let $h_{c_n}$ be the vector of internal neuron values of the nth child node of node $l$, and let $h_{c_n}^{\max} = max(h_{c_n})$, where $max$ is an activation function that returns a vector of all zeros except for the maximum value, e.g., $max([0.1, 0.8, 0.4]) = [0, 0.8, 0]$. Let $h_c^{\max}$ be the concatenation of all $N$ children node max values: $h_c^{\max} = [h_{c_0}^{\max,\top}, h_{c_1}^{\max,\top}, \dots h_{c_N}^{\max,\top}]^\top$, and let $M_l$ be the concatenation of the matrices lead from node $l$ to its children: $M_l = [M_{l,0}^\top, M_{l,1}^\top, \dots M_{l,N}^\top]^\top$. The update for a hidden node is

$$h_l = \frac{1}{Z}M_l^\top h_c^{\max}, \tag{16}$$

where $Z = N \parallel h_c^{\max} \parallel$. We show in the supplementary material this is equivalent to taking the weighted average of the conditional probability of child node assignments, where each conditional probability is weighted by the weighted average of the probability of its children values, which are a weighted probability of their children values, and so on (see Supplementary Note 2). Using this weighting thus conveys information about the probabilities of all descendants.

After signals propagate to the root/memory node, signals are propagated down the tree, and one-hot values assigned to each node. Let argmax be the activation function that assigns a one-hot to the maximum value, e.g., argmax([0.1, 0.8, 0.4]) = [0, 1, 0]. Let $h_l^*$ be the one-hot assignment for the $l$th node. At the memory node, the assignment is simply

$$h_L^* = \text{argmax}(h_L). \tag{17}$$

Signals are then propagated down according to

$$h_l^* = \text{argmax}(\lambda h_l + (1 - \lambda)M_{pa_l,l}h_{pa_l}^*), \tag{18}$$

where $pa_l$ is the parent node of $l$ and $\lambda$ a hyperparameter. For all recall tasks, we set $\lambda = 0.5$.

## Associative recall comparison
For our initial comparison to SoTA associative memory models, we tested our SQHN models on the same associative memory task as[19].

**Model architectures.** Each SQHN and MHN model had the same number of neurons at each of its nodes as there were images. Multi-level SQHN models had nodes with non-overlapping receptive fields. The SQHN L2 had local receptive field sizes $4 \times 4$ for CIFAR and $8 \times 8$ for tiny image net at its bottom layer, and a full receptive field of $4 \times 4$ at its top layer for both data sets. SQHN L3 had kernel sizes $2 \times 2$ and $4 \times 4$ for CIFAR and tiny image net, respectively, at its bottom layer. It had kernel size $4 \times 4$ at both its second and third layers for both data sets. The single-layer SQHN and MHN models we train have a single hidden layer where each neuron has full receptive fields.

**Hyperparameters.** For the MHN hyperparameters were not needed, since each image (or image patch) was stored in a separate column of

the matrix. In SQHN models $\alpha = \infty$ so that each image was stored at hidden layers using a separate set of neurons.

**Experiment setup.** In the moderate corruption task, 1024 images are presented. In the high corruption task, 128 are presented. The auto-associative task used white noise corruption with variances of 0.2 and 0.8 for moderate and high corruption task, respectively. For hetero-associative tasks, 25 and 75% of pixels are masked for moderate and high tasks. Masked pixels are treated as missing in these tasks and are therefore ignored by the bottom hidden layer of SQHN (rather than being treated as 0 values, which affects the normalization term). MSE is only computed in hetero-associative tasks between the output and the original pixels that were missing in the input.

Yoo et al.[19] tests the generative predictive coding network (GPCN) of[48], which trains offline. Yoo et al. develops a version for online training, called BayesPCN, and BayesPCN with forgetting, which prevents learning from slowing too much.

We also compared to MHN models. Following previous works (e.g., refs. 19,52, we update MHNs with a simple batch update where all the input vectors are stored in columns of a "memory matrix" $M$ and we set the temperature, $\beta$ to 10,000, essentially treating it as a argmax operator. The original model of ref. 45 performs retrieval via the operation

$$x^{(\text{new})} = M\,\text{softmax}(\beta M^\top \tilde{x}), \tag{19}$$

where $\beta$ is the temperature. Millidge et al.[52] showed this recall process is a kind of nearest neighbor operation, where a weighted average of memory vectors are returned, and memory vectors more similar to the input, according to a dot product measure, are given more weight. Recall is typically best when $\beta$ is large, and the nearest memory vector is returned. Millidge showed that other similarity measures work better than dot products. We show the best-performing model of Millidge et al. which uses the Manhattan distance. Finally, we also report the results of the gradient-based MHN (MHN-grad) listed in[19] that performs recall via a gradient-based inference procedure instead of the one-shot recall process shown above.

An important caveat is that the results reported by ref. 19 used images normalized to −1 and 1. Our SQHN model was designed for non-normalized images with values 0 to 1. To make the comparison fair, we multiplied the MSE scores for the SQHN by 4 since normalized images (ranging from −1 to 1) have an error range twice as large as unnormalized images (ranging from 0 to 1), which is then squared in the squared error: let $e = x - x^{\text{new}}$ be the unnormalized image and output error. If the image and output were normalized before computing the error we would be $2e$. If this error is then squared we get $4e^2$. Further, the standard deviation must be multiplied by two.

## Online-continual auto-associative memory tests (Fig. 2)
**Experiment setup.** In the online-continual learning task, images were presented online (one at a time and only a single pass through the data) in a non-i.i.d. fashion. In particular, images from the first task are presented online, then images from the second task are presented online, and so on. We tested grouping images by class (online class incremental or OCI) and by visually distinct data sets (online domain incremental or ODI). Each class and domain group had an equal number of images. In the ODI setting, the one-layer models trained one four different data sets/domains: MNIST, MNIST with flipped pixels, FMNIST, and FMNIST with flipped pixels. One-layer models were tested with 300, 1300, and 2300 hidden layer neurons on MNIST data sets, and 700, 2000, and 3300 on CIFAR-100. The three-layer models trained on CIFAR-10 with dark pixels ($x^{\text{dark}} = x \times 0.5$), CIFAR-10 with bright flipped pixels ($x^{\text{light}} = (-1 \times x + 1) \times 0.5 + 0.5$), SVHN dataset with dark pixels, and SVHN with bright flipped pixels.

**Model architectures.** The three-layer SQHN model had nodes with non-overlapping receptive fields. Receptive fields for each layer were $4 \times 4$, $4 \times 4$, and $2 \times 2$, at the first, second, and third hidden layers, respectively. The one-layer model was tested with node sizes 300, 1300, and 2300. The three-layer model was tested tested with node sizes 200, 600, 1000. During test time, training images with a noise variance of 0.2 are presented. The SQHN is compared to the modern Hopfield network (MHN)[44,45].

We compare to the MHN of ref. 45, in particular, because it is the most similar architecturally to SQHN, and has a very high capacity relative to other MHNs[45]. The one-layer model performs recall as:

$$x_{\text{new}} = M \, \text{softmax}(\beta M^T x), \tag{20}$$

where $x$ is the input vector and M with weight matrix, and $\beta$ a scalar temperature. This same mechanism can be stacked into tree-like hierarchies as well, similar to the multi-level SQHN. For the multi-level MHN we essentially use the same architecture (same local receptive field and node sizes) but replace the argmax with softmax and remove the normalization.

One issue we ran into was there was no standard way to train this model online for auto-associative tasks like the ones used here. The original model of ref. 45 was trained with BP, but not used for auto-associative memory. Instead, it was treated like the attention layer of a transformer, where $M$ is generated by a separate set of weights. Thus, we train the model like the original MHNs of ref. 44, which were used for auto-association, where BP is used to directly optimize $M$. Since we are training to reduce recall MSE, we directly optimize MHNs to reduce recall MSE. We test the model trained with plain SGD, SGD with Adam optimization[61], EWC++[53] which is an online version of EWC[35], and episodic replay with a small memory buffer[40]. EWC++ uses a moving average of the Fisher information matrix, $F$, to modulate parameter updates. Parameter updates for EWC are regularized loss gradients where the objective, $\bar{L}$ is:

$$\bar{L}^k = L^k + \lambda \sum_i F_{\theta_i^{k-1}} (\theta_i - \theta_i^{k-1})^2, \tag{21}$$

where $\theta_i^{k-1}$ is the ith value of the parameters from task $k-1$ and $\lambda$ weight the regularization. In EWC++, the Fisher is computed as a moving average $F^t = \alpha F^t + (1-\alpha)F^{t-1}$, where $t$ is the current training iteration, and a moving average of parameters is kept as well. At the end of each task, $F_{\theta^{k-1}}$ is set equal to $F^t$ and the moving average of parameters set equal to $\theta_{k-1}$. One issue is that EWC++ needs to know task boundaries, so it can store the parameters and Fisher matrix from previous task. In our task, no such supervision is allowed. To deal with this, we just treat each data point as its own task and compute a moving average of the parameters, the same as the Fisher information matrix. This should still allow parameters and Fishers to maintain information about previous tasks. The $\lambda$ and $\alpha$ hyper-params are found via grid search. Generally, we find that regularization provides little to no benefit, because the training runs are short and therefore training speed is as important as preventing forgetting. EWC slowed down training speed, too when $\lambda$ was significant, so the regularization provided little help.

Episodic replay with a tiny memory buffer[40] stores a small sample of previously observed data-points, then uses SGD to update parameters each iteration using the whole sample as a mini-batch. Despite using a small buffer, this approach has shown to be highly effective in continual classification. Methods that sample a mini-batch from a larger buffer can work better[16]. However, we are interested in memory and compute-efficient algorithms and are using relatively small data sets. Thus, we use the tiny memory buffer method. There are several algorithms for deciding when to store a data point in the buffer. We use the reservoir sampling method (see ref. 40), which is a simple method

that has shown to consistently be better than other methods on classification tasks. The reservoir method is simple: let $n$ be the maximum number of data point that can be stored in the buffer, and $t$ be the total number of data points observed. For the data point, $x^t$, at each iteration $t$, check if the buffer is full. If it is not full, add the data point to the buffer. If the buffer is full, randomly overwrite one data point in the buffer with $x^t$, with probability $\frac{n}{t+1}$.

**Hyperparameters.** Specific hyperparameter settings for each model can be found in the code. The hyperparameters were found via grid searches. The SQHN model's only hyperparameter is the $\alpha$ parameter which controls neuron growth. Ten to fifteen values between 1000 and 100,000 were tested, and the best-performing one was used. The MHN model had both the inverse temperature, $\beta$, and the learning rate hyperparameter. For beta we search 10–15 values between 0.01 and 100 for $\beta$ and between 0.001 and 0.9 for the learning rates.

## Noisy encoding
**Experiment setup.** We compared one and three hidden layers SQHN and MHN models on EMNIST and CIFAR-100, respectively, in the noisy encoding task. The single-layer models were tested with node size 300, and were presented with 300 images. Trees had a node size of 150 and were presented with 150 images. Therefore, any inability to recall images must be due to an inability to remove noise during learning, rather than capacity limitations. Images were either Gaussian samples (white noise added then clamped to range 0 to 1) or binary samples, were drawn from the original image values. For each image, some number of samples were drawn and presented to the models online in a sequence, before moving to the next image. For EMNIST images, we test 1, 5, 20, and 50 samples. For CIFAR-100, we tested 1, 10, and 20 samples. At no point during training was the original, non-corrupted image presented to the model. At test time, the original, non-corrupted image was presented, and the model had to be reconstructed. We also tested an SQHN model with a slight alteration (SQHN+), where after the hidden states are computed for the first image sample, the hidden states are held fixed for the remainder of the samples. This ensured that samples from the same image were encoded to the same latent state. Although SQHN was able to do this well on its own, it sometimes mapped samples from the same image to different latent state in the high-noise, binary sample scenario. In these cases, SQHN+ performed better.

**Model architectures and hyperparameters.** The same architectures and hyperparameter searches were used as in the last online auto-association task.

## Episodic recognition
**Model architectures.** For this test we used single hidden layer SQHN and MHN architectures with 300 neurons at the hidden layer.

**Experiment set-up.** For the episodic memory comparison between SQHN and MHN, we use a one-layer model and MNIST data sets. The train set is pulled from the MNIST training data. The in-distribution hold-out set is from MNIST test data. The out-of-distribution set is from F-MNIST. We test two methods for performing recognition in an MHN with one hidden layer. The first method uses the activities at the hidden layer as a measure of familiarity/similarity. If that value is above a threshold, $\rho$, then the model judges it has observed the data point. The second method keeps a moving average, $\mu$, of the recall MSE during training, and uses a scalar multiple of this average, $\rho\mu$, as the decision threshold. All hyperparameters were found with grid search. Learning rates are updated to achieve the best performance in terms of recall MSE. $\rho$ updated to achieve the best recognition accuracy. 3000 data points from EMNIST are presented in an online and i.d.d. fashion. At each test point, all previously observed train data are presented

along with an equal number of in-distribution and out-of-distribution data. We plot the MSEs of MHN and SQHN for each data set to show that MHN have no performance difference between training and in-distribution data, making it good at generalization but unable to perform recognition.

**Hyperparameters.** A grid search was used to find the $\beta$ and learning rate values that achieved the best MSE after 3000 iteration for the MHN. The same procedure was used to find the best $\alpha$ value for SQHN models. A further grid search was used for $\rho$ testing 10–15 values between 0 and 1.

**SQHN architecture compare (Fig. 5B, D, E)**
**Auto-association with varying amounts of corruption (Fig. 5B).** All SQHN models had 1000 neurons at each node and are trained to memorize 1000 images. SQHNs are trained so that each image is encoded in a unique set of one-hot vectors at hidden nodes, (which can be done in practice by setting $\alpha$ to a very large number). This means that all SQHN models are not over capacity, and an inability to recall images is due only to their inability to handle varying amounts or types of corruption. Importantly, only auto-associative, and not hetero-associative, recall is tested. The masks added to images are treated as corrupted pixels (i.e., their values are taken as input) rather than missing pixels. For noise tasks, we add white noise to the images and then clamp the images to values between 0 and 1. Noise variances tested are [0, 0.05, 0.15, 0.25, 0.4, 0.5, 0.75, 1, 1.25, 1.5]. Fractions masked that are tested are [0, 0.1, 0.25, 0.5, 0.75, 0.875, 0.9375]. For masking tasks, a rectangle with random length and width (less than or equal to the width and height of the image) is sampled, and its position within the image is randomly sampled. The black mask sets pixels equal to 0. The color mask randomly selects an RGB value from a uniform distribution, and sets pixels in the mask equal to that value. Noise mask sets mask values equal to a white noise sample (variance 1), clamped to values between 0 and 1. The one-level SQHN L1 model has an input layer kernel size equal to the size of the image. SQHN L1 essentially performs nearest neighbor operations comparing each images to the set of training images, which are stored in columns of its weight matrix, using mean-shifted cosine similarity. SQHN L2 has an input layer kernel size of $8 \times 8$, $16 \times 16$, and $32 \times 32$ for CIFAR-10, Tiny Imagenet and Caltech 256, respectively. The second layer kernel across all data sets is sized $4 \times 4$. SQHN L3 has an input layer kernel size of $4 \times 4$, $8 \times 8$, and $16 \times 16$ for CIFAR-10, Tiny Imagenet and Caltech 256, respectively. The second and third layer kernel sizes across all data sets are sized $4 \times 4$.

**Online auto-associative learning (Fig. 5D).** For this task, one, two, and three hidden layers models were trained in the online i.i.d. scenario on CIFAR-100. Models were trained with 200, 600, and 1000 neurons at hidden nodes. SQHN L1 model had an input layer kernel size equal to the dimension of the image ($32 \times 32$). SQHN L2 had an input layer kernel size of $4 \times 4$ (which worked better than the $8 \times 8$ kernel used in the experiment above), and a second layer kernel size of $8 \times 8$. SQHN L3 has an input layer kernel size of $4 \times 4$, $8 \times 8$, and $16 \times 16$ for CIFAR-10, Tiny Imagenet and Caltech 256, respectively. The second and third layer kernel sizes across all data sets are sized $4 \times 4$. $\alpha$ was set very large, so networks memorized the first J data points, up until capacity was reached. We measured recall accuracy without any corruption, and noticed L2 and L3 architectures performed much better than L1. We suspected this was because the L2 and L3 architectures learn the representation of small features that are generalized widely across training sets. To test this we also measured test MSE on the test set from CIFAR-100.

**SQHN episodic recognition comparison (Fig. 5E).** For the recognition task, a train set, in-distribution hold-out set, and out-of-distribution set are needed. We use CIFAR-10 training images for the

train set, a hold-out/test set of CIFAR-10 images as the in-distribution set, and CIFAR-100 with flipped pixels as out-of-distribution set (i.e., each image is multiple by −1 then 1 is added). SQHN L1 is given 500 neurons at its hidden node. SQHN L2 and L3 are given 500 neurons at their memory node. Since only the memory node is used directly for recognition, we pre-train the lower layers of SQHN L2 and L3 for 1000 iterations on CIFAR-10 images. SQHN L2 has kernel sizes $4 \times 4$ and $8 \times 8$ at hidden layers, and a channel size of 40 at the first hidden layer. SQHN L3 has kernel sizes $2 \times 2$, $4 \times 4$, and $4 \times 4$ at the first, second, and third hidden layers, respectively, and channel sizes of 40 and 200 at the first and second hidden layers. After pre-training during the training phase, images from the train set are presented online in and i.i.d. manner. Only the weights leading into the memory node are updated. During testing, models are presented with a set of images, which are composed of all of the training images observed so far, an equal number of in-distribution images, and an equal number of out-of-distribution images. The best guessing strategy is to guess all data points are new which yields 66% accuracy.

## Reporting summary
Further information on research design is available in the Nature Portfolio Reporting Summary linked to this article.

## Data availability
All image data sets used in this paper are publicly available for download. TinyImageNet data set can be downloaded at the following link: https://www.kaggle.com/c/tiny-imagenet. The remaining data sets are automatically downloaded through pytorch dataloaders, the code for which can be found in our open-source repository (see next section).

## Code availability
The code can be found at https://github.com/nalonso2/SQHN. Code was written in Python 3.7.6 using Pytorch version 1.10.0 to implement models and perform differentiation for models that use backpropagation.

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

## Acknowledgements

We would like to thank Mark Steyvers for the illuminating discussions we had on episodic memory and novelty detection. This work was supported by the National Science Foundation Grant IIS-1813785 and the Air Force Office of Scientific Research Grant FA9550-19-1-0306.

## Author contributions

Nicholas Alonso and Jeffrey Krichmar conceived the SQHN model design and experiment designs. Nicholas Alonso developed the mathematical theory and results behind the SQHN network, wrote the code for the simulations, and was the lead writer of the manuscript. Jeffrey Krichmar advised and edited the writing of the manuscript.

## Competing interests

The authors declare no competing interests.
