## [Peer Review File · Nature Communications]

REVIEWER COMMENTS

Reviewer #1 (Remarks to the Author):

This work introduces a novel kind of hierarchical, sparse, and discrete (quantised) Hopfield network, with a tree structure. The authors claim that other shapes are allowed, but they mainly explore this one, with 1,2,or 3 layers. The main task they tackle is that of associative memories, popularised in the 80's thanks to Hopfield's works, which is now living a second 'golden era' thanks to continuous state Hopfield Networks, and their connections to transformer models. A parallel line of research is that of performing associative memories using neuroscience-inspired models, such as predictive coding models [1,2], that also perform a kind of MAP computation. This work is nicely placed at the intersection of the two areas, as it presents strong similarities with both models. In fact, Eq.4 really is the update rule of predictive coding, up to a weighting constant also introduced in a new work [3].

First point: In a revised manuscript, I would like to see a discussion of the similarities, and differences, between your proposed model, and both Hopfield Networks and predictive coding models. What are the drawbacks of both, and why your hybrid approach is a solution? I believe that adding 2/3 paragraphs explaining this would improve the quality of the manuscript, as well as making the objective of the paper crystal clear. More generally, a related works section is lacking. I believe it should be added, to better place this work in the literature. For Hopfield Networks, the "HN is all you need" paper has a good literature review to take inspiration from. For PC, Friston's "Does predictive coding have a future?"[4] is a good reference.

Methods:

The section explaining the methodology and the relevant equations is sound and clear. As minor comments, I would provide the pseudocode in the main body, as well as dividing the "learning" sub-sub-section in two different parts, that is "parameter update", and "structure learning" (the process of adding additional neurons when needed). In fact, I believe that paragraph to be interesting, and would like it to be separated and highlighted. I would also delegate Section 2.2.4 to a later stage.

Experiments: ^{SEP}The experimental section is long and sound, and shows the improved performance of this method against the baselines considered. No important baseline has been overlooked, and the results clearly show that the method of the authors does take the best of the both worlds previously mentioned. I would, however, ask the authors to perform two simple experiments, that I believe would make the contribution stronger.

Second point: suggestions on experiments:

1) Classification tasks: When asking the model to memorize images, the proposed model is able to add extra neurons if it finds the image novel. However, in multiple experiments, there is a threshold on how many neurons can be added. If the capacity of the model is lower than the number of the stored images, is the model able to compress information, and hence generalize to unseen patterns? An interesting experiment to show, would be to perform image classification on the MNIST dataset by adding an extra layer with 10 neurons (or, consider 10 root nodes) on top of the hierarchy, and ask the model to memorize (a subset of) the MNIST dataset as follows: every time a '3' is shown, we can fix the neuron related to the third label to 1, and ask the model to retrieve the image. At test time, we can present an unseen 3 from the test set, and perform a feedforward step. I am aware that classification is not the main goal of the work, but some Hopfield Networks have been shown to perform well on classification tasks (Dense associative memories), and hence it would be nice to show whether this model is also able to

Before proceeding to the next point, I have a question for the authors: does the model always output an existing memory, regardless of the noise? This is generally the case for Hopfield Networks (especially when using high precisions beta parameters). If the output of the model is always an existing memory, it would be nice to be able to discriminate how much such output can be trusted: is a specific image returned because it is actually similar to the cue, or simply because the model had to produce an output? If this is the case, it would be nice to add a discussion on it, and maybe incorporate the following:

2) Corrupted episodic memory. The proposed model can be queried to state whether it has already seen an image, or not. It would be nice to go a little further than that: in some applications, it would be useful to test whether the model has already seen a 'corrupted' image, or not. The experiment could be performed as follows: after having trained a SQHN on a dataset of natural images (say, CIFAR10), we want to gather two extra datasets: a corrupted version of the training set (e.g., with white noise), and a corrupted version of the test set. Is the model able to recall whether it has already seen a non-corrupted version of the proposed, corrupted, image?

On reproducibility:

I believe that the manuscript can be improved in this regard. I would like to see a detailed explanation on how all the experiments have been performed (all hyper parameters used, which grid search, exact architecture for every experiment, number of parameters, initialisation, etc...). As the manuscript stands

now, I would not be able to reproduce the results presented. Note that this is not a judgment on the soundness of this work, as I trust all the results presented. It is only for the sake of reproducibility.

Minors:

- The structuring of the sections should be changed: Sec. 2 is huge, and I believe should be divided in multiple sections.
- Section 2.2.4 seems out of place there. Maybe move it in the experimental section?
- Add a section on growth/structure learning (after 2.2.3).
- The citations below are suggestions of missing ones, that should be incorporated in the manuscript.

[1] Tang, Mufeng, et al. "Recurrent predictive coding models for associative memory employing covariance learning." *PLoS computational biology* 19.4 (2023): e1010719.

[2] Tang, Mufeng, Helen Barron, and Rafal Bogacz. "Sequential Memory with Temporal Predictive Coding." *arXiv preprint arXiv:2305.11982* (2023).

[3] Alonso, Nick, Jeff Krichmar, and Emre Neftci. "Understanding and Improving Optimization in Predictive Coding Networks." *arXiv preprint arXiv:2305.13562* (2023).

[4] Friston, Karl. "Does predictive coding have a future?." *Nature neuroscience* 21.8 (2018): 1019-1021.

Reviewer #3 (Remarks to the Author):

The authors would likely need to compress this work to fit within the word count for this venue.

This seems to be interesting work, I'd recommend the authors make a few clarifying modifications. For example, in the introduction they use the term quantization, which is an overloaded term (e.g., a lot of work on 8-bit quantization for computing with reduced numerical precision). However, this work effectively maps on to vector quantization. While this is obvious from context, something earlier on that states this would be useful.

There's something counterintuitive going on here where I would imagine that sparsity should reduce a Hopfield network's capacity, since it could be counter to compression. Is there some explanation for what the authors have noticed here? I would also appreciate some more details about how the model's capacity is affected through the various choices being made?

Would it be fair to say that the described algorithm is similar to belief propagation, in how the local updates are implemented? That would be a classical method to do inference on probabilistic graphical models of the type described. Would the authors find that to be a useful reference to cite? Would the authors have any comparison to using BP or any other standard approach here?

Point-by-Point Response to Reviewer Comments

We thank the reviewers for taking the time to review our manuscript and for providing useful suggestions and interesting questions. We have done our best to address each of the suggestions and to answer each question. Our responses, which provide summaries of revisions, can be found below. More details about revisions can be found in the revised manuscript. Reviewer comments are quoted and italicized. Our responses are shown in red.

Reviewer #1

“First point: In a revised manuscript, I would like to see a discussion of the similarities, and differences, between your proposed model, and both Hopfield Networks and predictive coding models. What are the drawbacks of both, and why your hybrid approach is a solution? I believe that adding 2/3 paragraphs explaining this would improve the quality of the manuscript, as well as making the objective of the paper crystal clear. More generally, a related works section is lacking. I believe it should be added, to better place this work in the literature. For Hopfield Networks, the “HN is all you need” paper has a good literature review to take inspiration from. For PC, Friston’s “Does predictive coding have a future?”[4] is a good reference.”

We agree a good review and comparison to previous literature is necessary for this work. Due to word limitations, in the original manuscript we provided an extended lit review in the supplementary materials rather than the main body, which included a comparison to modern hop nets. We did not have a section on PC, however, and agree discussing relations between the two is important.

In the revised version, we added a lit review to the main body after the description of the model, including a discussion of relations between SQHNs and similar sparse neural nets, online-continual learning approaches, modern Hop Nets, and PCNs. If the paper is accepted we will work with the editor to meet word count limitations.

“Methods: The section explaining the methodology and the relevant equations is sound and clear. As minor comments, I would provide the pseudocode in the main body...”

In the revised work, we added pseudo-code for the inference and learning algorithm to the main body in the model section (which is section 2 in the revised version).

“...as well as dividing the “learning” sub-sub-section in two different parts, that is “parameter update”, and “structure learning” (the process of adding additional neurons when needed). In fact, I believe that paragraph to be interesting, and would like it to be separated and highlighted.”

We agree it would be useful to distinguish between parameter updates and neuron growth. As recommended, we split the learning sub-sub-section in two. We titled the first ‘Parameter Updates’ and the second ‘Parameter Growth’, instead of structure learning to avoid confusion with techniques for learning the structure of the graph itself, which we do not do here.

“I would also delegate Section 2.2.4 to a later stage.”

We agree moving this section to a later stage might make more sense, since this section is so specific to just the episodic memory task. We moved section 2.2.4 to the later experimental section on the episodic memory task.

“Experiments: The experimental section is long and sound, and shows the improved performance of this method against the baselines considered. No important baseline has been overlooked, and the results clearly show that the method of the authors does take the best of the both worlds previously mentioned. I would, however, ask the authors to perform two simple experiments, that I believe would make the contribution stronger. Second point: suggestions on experiments: 1) Classification tasks: When asking the model to memorize images, the proposed model is able to add extra neurons if it finds the image novel. However, in multiple experiments, there is a threshold on how many neurons can be added. If the capacity of the model is lower than the number of the stored images, is the model able to compress information, and hence generalize to unseen patterns? An interesting experiment to show, would be to perform image classification on the MNIST dataset by adding an extra layer with 10 neurons (or, consider 10 root nodes) on top of the hierarchy, and ask the model to memorize (a subset of) the MNIST dataset as follows: every time a ‘3’ is shown, we can fix the neuron related to the third label to 1, and ask the model to retrieve the image. At test time, we can present an unseen 3 from the test set, and perform a feedforward step. I am aware that classification is not the main goal of the work, but some Hopfield Networks have been shown to perform well on classification tasks (Dense associative memories), and hence it would be nice to show whether this model is also able to.

While we did provide some preliminary results on generalization abilities in image reconstruction (see figure 5D,) we agree that generalization abilities of SQHN requires further testing and focusing on generalization abilities, especially through classification tasks, would be a great next step. However, the main goal of the paper was to test online and continual memory encoding, and as such classification is outside of the scope of the main paper. We do mention in the last paragraph of the discussion section that we plan, in future work, to further test the generalization abilities of SQHNs, and we say that classification could be one way to test the generalization abilities of the network.

“Before proceeding to the next point, I have a question for the authors: does the model always output an existing memory, regardless of the noise? This is generally the case for Hopfield Networks (especially when using high precisions beta parameters).”

It depends on what is meant by an existing memory. The recall process of a one layer SQHN works similarly to a modern Hop Net with precision set to infinite. A one layer SQHN stores a proto-type/memory vector in each column of its matrix and returns the column vector that is most similar to the input, where similarity is measured via a mean shifted cosine similarity (see methods), whereas modern Hop Net uses dot product to measure similarity (although this idea has been extended by Millidge et al. (2022) to the universal Hopfield network which encompasses any similarity measure). If we consider each column vector to be a memory, then,

yes, a one layer SQHN always returns an existing memory. However, unlike modern Hop nets (or universal Hop nets), it is important to remember that in SQHNs these memory vectors are often (especially after capacity is reached) averaged across multiple vectors. In these cases, the returned memory vector is not a particular data point, but instead a proto-type/avg of similar data points. A one level SQHN is therefore better viewed as a dynamic, online K-means model that uses mean-shifted cosine sim for the similarity measure.

However, this interpretation does not fully apply to a multi-level SQHN architecture. In these networks at lower layers, nodes in the network only take in a small local patch from the layer below. Therefore, when such a network returns an image, the bottom layer is actually outputting a set of sub-vectors (one for each image patch). These sub-vectors can be memorized from one previous data point but are often (especially after capacity is reached) averaged across patches from multiple previous data points. Representations of small image patches can often be reused quite well across multiple training images, which helps (we found empirically) with recall capacity and generalization to unseen images. With new images novel combinations of memory vectors may be outputted for each image patch. Thus, in the multi-level tree architectures two different memorized images may utilize some of the same image patch proto-types, and new outputs may be produced given new images.

One more thing to note here is that although during our memory tests we force the network to output a memory vector for each image patch, during training the network is not forced to output anything. Nodes that have a neuron value that is larger than the growth threshold will output a memory vector. However, nodes that do not have a value higher than the threshold will not index/output an existing memory vector but will instead store a new one. In other words, when there is no value above the threshold, this network infers there is no memory of the sort of input it is receiving, i.e., the input is a new kind of input and should belong to a new memory vector.

Because multi-level SQHNs often use overlapping codes between images, in the paper we find it easiest to describe memories as stored in discrete neural codes that are at maxima of the energy landscape. The higher the energy, the more accurate the recall process will be. We summarize this idea in figure 1 and section 2.2.2. Also section 2.4 discusses why a tree like architecture is useful for generalization.

In the revised manuscript, we added several sentences to section 2.2.2 explaining some of these points. We hope this answers your question. Let us know if you or the paper needs further clarification.

If the output of the model is always an existing memory, it would be nice to be able to discriminate how much such output can be trusted: is a specific image returned because it is actually similar to the cue, or simply because the model had to produce an output? If this is the case, it would be nice to add a discussion on it, and maybe incorporate the following: 2) Corrupted episodic memory. The proposed model can be queried to state whether it has already seen an image, or not. It would be nice to go a little further than that: in some applications, it

would be useful to test whether the model has already seen a 'corrupted' image, or not. The experiment could be performed as follows: after having trained a SQHN on a dataset of natural images (say, CIFAR10), we want to gather two extra datasets: a corrupted version of the training set (e.g., with white noise), and a corrupted version of the test set. Is the model able to recall whether it has already seen a non-corrupted version of the proposed, corrupted, image?"

We like the idea of this test too. We ran this test for SQHN architectures with 1, 2, and 3 hidden layers on CIFAR10. See figure 5E in the new manuscript. We found that all models performed a bit worse with noise than without. Further, the architectures with more hidden layers, which were more sensitive to noise than the one hidden layer model on auto-associative recall, performed worse on the episodic memory task as well. These results should not be too surprising, given these models performed worse at recall to under noise. And it is intuitively consistent with biology as it is likely harder for animals to recognize stimuli in noisy environments than non-noisy environments too. We are not sure if this answers your initial question. Let us know if you have further questions on this.

"On reproducibility: I believe that the manuscript can be improved in this regard. I would like to see a detailed explanation on how all the experiments have been performed (all hyper parameters used, which grid search, exact architecture for every experiment, number of parameters, initialisation, etc...). As the manuscript stands now, I would not be able to reproduce the results presented. Note that this is not a judgment on the soundness of this work, as I trust all the results presented. It is only for the sake of reproducibility."

We agree we could add more in this regard.

In the revised manuscript methods section, we could not add all of the hyperparameters grid searched over (this is hundreds of numbers), but we did add a description of the range of values we ran grid searches over and how we ran the grid searches. We made separate labeled paragraphs (labeled 'hyperparameters') for this information. We also made labeled paragraphs for architecture descriptions (labeled 'Model Architectures') and added detail where needed. Note that our code will be open sourced upon publication, and all hyperparameters used in the simulations in the paper are set as default values for each simulation. The code is set up so each simulation, with used hyperparams can be run with a single command, and plots can be easily reproduced. Open sourcing will also provide more detail on the model architecture.

"Minors:

- The structuring of the sections should be changed: Sec. 2 is huge, and I believe should be divided in multiple sections."

We agree section 2 is quite large.

In the revised version we now have the following sections: 1) Introduction, 2) Model, 3) Related Works, 4) Results, 5) Discussion. If accepted we will work with the editor to find the best sectioning of the paper.

"- Section 2.2.4 seems out of place there. Maybe move it in the experimental section?"

Agreed. We complied with this request, see above.

“- Add a section on growth/structure learning (after 2.2.3).”

Agreed. We complied with this request, see above.

“- The citations below are suggestions of missing ones, that should be incorporated in the manuscript.”

Agreed. These citations are added and incorporated into the new section on predictive coding. See comment on predictive coding section above.

[1] Tang, Mufeng, et al. "Recurrent predictive coding models for associative memory employing covariance learning." *PLoS computational biology* 19.4 (2023): e1010719.

[2] Tang, Mufeng, Helen Barron, and Rafal Bogacz. "Sequential Memory with Temporal Predictive Coding." *arXiv preprint arXiv:2305.11982* (2023).

[3] Alonso, Nick, Jeff Krichmar, and Emre Neftci. "Understanding and Improving Optimization in Predictive Coding Networks." *arXiv preprint arXiv:2305.13562* (2023).

[4] Friston, Karl. "Does predictive coding have a future?." *Nature neuroscience* 21.8 (2018): 1019-1021.

Reviewer #3:

“The authors would likely need to compress this work to fit within the word count for this venue.”

Our original manuscript was ~5700 words. Currently, our revised manuscript after addressing reviewer comments is ~6500 words. We understand this is over the recommended word count of 5000 words. If accepted, we will ask the editor if the paper needs compressing, and work with the editor to compress the paper if it does.

“This seems to be interesting work, I'd recommend the authors make a few clarifying modifications. For example, in the introduction they use the term quantization, which is an overloaded term (e.g., a lot of work on 8-bit quantization for computing with reduced numerical precision). However, this work effectively maps on to vector quantization. While this is obvious from context, something earlier on that states this would be useful.”

We agree this should be clarified. In the revised manuscript, we made clearer in section 2.1, where quantization is first discussed, that we are proposing to use quantized neural codes rather than parameter quantization, and that we are proposing to use this quantized neural code to perform a general kind of vector quantization process.

“There's something counterintuitive going on here where I would imagine that sparsity should reduce a Hopfield network's capacity, since it could be counter to compression. Is there some explanation for what the authors have noticed here?”

We agree this is an important question. To start, let's compare a simple, one-layer SQHN to the continuous modern Hopfield network, which is most similar to the one-layer SQHN. Whereas the SQHN uses a sparse one-hot at its hidden layer, which indexes the one memory vector with the highest similarity to the input (measured by mean-shifted cosine similarity, see methods), the modern Hop net uses a softmax, which gives some weight to every memory vector, where the weight depends on similarity to the input (measured by the dot product). Let's also assume the ideal case where there is one training vector stored in each column/memory vector of the weight matrix.

Let's first consider the case where input is not corrupted. For a simple one hidden layer SQHN, we showed in theorem 1 the capacity, under minimal assumptions, is N , or the number of hidden neurons. Modern Hop nets have the same capacity (see Ramsaur et al. 2022, theorem 3), except under more complex and limiting assumptions about input dimension, normalization of input vectors, and all sorts of hyperparameter settings, which are assumptions we do not have to make. What accounts for this? SQHNs always return the memory vector most similar to the input measure by mean-shifted cosine sim, and the outcome of this process simply does not depend on the input dimension, hyper params, or norms of input vectors (in the uncorrupted case), whereas modern Hop Nets return a weighted average of memory vectors, and these weightings depend on the dimension of the inputs, their scaling, and hyper-parameters, like temperature.

Now that being said, there is the point that returning some weighted average or linear combination of memory vectors should allow modern Hop nets to return (and possibly store) far more vectors than it has neurons at its hidden layer. Whereas the winner-take-all operation of the SQHN limits the number of possible outputs it can produce to N . This is a drawback of the winner-take-all operation. However, there are several things to say here 1) it does not seem the ability to return arbitrary linear combinations of memory vectors provides similar modern Hopfield nets with that much more capacity in practice or in theory, according to Ramsauer et al., than the SQHN 2) The winner-take all operation has great benefits in the actual scenarios we are interested in using these networks on, which is associative recall given corrupted input (see below). Most formal analysis only considers uncorrupted cases. 3) In more complex tree-like architectures, the number of possible outputs and SQHN can produce are far more than the number of neurons at each node, e.g., consider a multi-level SQHN with 16 nodes at the bottom layer that each take input from a different square image patch, and assume each node has N neurons. This network can produce, in principle, N^{16} possible outputs.

In the case where input is corrupted, we claimed a quantized neural code help remove corruption from the input by ensuring (under reasonable assumptions) that corrupted versions of a stored training vector, x , are all mapped to the same hidden latent code, h^ , leading the network to output an exact reconstruction of the original x . In modern Hop Nets, which use*

softmax, each corrupted version of x will be mapped to a slightly different latent code that distributes some weight across all memory vectors.

One way to think about this difference is that SQHNs are doing something more akin to MAP inference: return the memory vector x which is most probable given network parameters and the corrupted input (return the best guess). The modern hop net on the other hand does something more akin to computing a posterior distribution, i.e., it returns a weighted average of the vectors taking into account uncertainty over which memory vector is the correct/best one. Accounting for uncertainty is useful in some applications, but it will be bad at removing noise, since the noise will add uncertainty (i.e., it will add more even weighting across memory vectors) thereby worsening the reconstruction. Consistent with classic work on noisy decoding processes, we think MAP inference is a good framework for auto-associative tasks. Although this is a simple and idealized example, we proposed and supported empirically that these properties scale to more complex architectures, like the multi-level tree SQHN architectures we tested.

In sum, while a sparse MAP-like code does limit the number of outputs the model can produce compared to Hop Nets with dense codes, it makes the model far more robust to corrupted inputs, and does not seem to reduce capacity relative to similar Hop Nets. Further, the normalization SQHNs uses during inference ensure capacity is maintained under fewer assumptions than that for similar Hop nets, like continuous modern Hop nets.

In the revised draft, we have a more explicit comparison between SQHNs and Hop nets in a new related works section (section 3), and we summarize some of these same points in that section.

“I would also appreciate some more details about how the model's capacity is affected through the various choices being made?”

Given uncorrupted inputs a one hidden layer SQHN can store at least N training vectors (see theorem 1). However, the capacity of tree-like architectures is more interesting for two reasons. 1) As noted above, tree-like architectures can produce far more outputs than just N (it can produce N raised to the number of lower layer hidden nodes), whereas a single hidden layer SQHN can only produce N outputs. 2) As is well known, natural images are well described as compositions of many local features combined together. Thus, it seems possible a priori that a tree like architecture can store many more images than N , since the lowest layer represents the image as a composition of local features, with N possible values, and images in the train set may share many local features. We did not prove this formally (in part because it is very difficult to make general claims about arbitrary tree-like architectures, given their many possible variables, like number of nodes, receptive field size and shape, number of hidden layers, graph structure, etc.). However, we did find empirically that for a small but non-zero recall threshold, tree like architectures 1) were able to recall far more than N images, and significantly more than the single layer SQHN (see figure two bottom set of plots, and figure 5D), and 2) they tended to generalize better to new data points suggesting the representation of local features were reusable across many data points including new ones (figure 5D).

In the revised manuscript, we added several sentences to section 2.4 to make these points clearer. Section 2.4 is where we compare SQHN networks with different numbers of layers and nodes at each layer, and where we observe the networks with more layers have slower forget rates and better performance on certain tasks.

“Would it be fair to say that the described algorithm is similar to belief propagation, in how the local updates are implemented? That would be a classical method to do inference on probabilistic graphical models of the type described.”

We agree this is an important comparison. If by ‘belief propagation’ you are referring to the sum-product algorithm, there are some differences between the SQHN inference procedure and sum-product algorithm. For one, the sum-product algorithm computes the marginal distribution for each node, whereas the SQHN computes something more akin to the MAP value for each node. In this way, the SQHN algorithm is more similar to the max-product algorithm that performs MAP inference for belief nets than the sum-product.

We discuss the max-product algorithm and compare it to SQHNs in the supplementary materials. In the revised manuscript, this section is supplementary B.2.

“Would the authors find that to be a useful reference to cite?”

We agree this would be a useful reference to cite. In the original manuscript, we discussed the max-product algorithm for belief networks in the supplementals and cited textbook references.

In the revised manuscript, we added several more references to section B.2 and section 2.2.2 to be more thorough.

“Would the authors have any comparison to using BP or any other standard approach here?”

We provided a detailed formal description of the differences between the max-product algorithm and our inference algorithm, as well as its motivations, in the supplementary materials, section B.5. In sum, the operations of SQHN inference are similar, but different in certain ways: 1) unlike max-product, the SQHN procedure only involves standard neural network operations (i.e., matrix multiplies + non-linearities) and 2) the motivation for which is that the SQHN inference procedure is much more straight-forward to implement and better utilizes hardware which the bio-inspired machine learning community typically uses (e.g., GPUs in the machine learning community and memristors in the neuromorphic community), all while achieving high performance.

As far as empirical/performance comparisons, we aimed to compare our SQHN to SOT models for auto-association and online-continual learning. We do not know of any recent work that uses classic belief prop algorithms to achieve anything close to SOT results on these tasks, so we did not compare the SQHN to belief prop methods in our paper.

In the revised manuscript we added a high-level description of the similarities and differences between max-product and SQHN procedure to section 2.2.2. We also added a reference, in section 2.2.2, to supplementary section B.2 so that readers interested in the relations between SQHN inference and the max-product algorithms can easily find more details.

REVIEWERS' COMMENTS

Reviewer #1 (Remarks to the Author):

I thank the reviewers for their effort in addressing my many comments. I'm happy and satisfied with both the rebuttals and revised manuscript.

Response to Reviewer Comments

We only received comments from reviewer 1. Our response is below in red.

Reviewer #1 (Remarks to the Author):

I thank the reviewers for their effort in addressing my many comments. I'm happy and satisfied with both the rebuttals and revised manuscript.

We thank reviewer 1 for their helpful comments and are glad our revisions were found satisfactory.